

# Relaxation of the order-parameter statistics
# in the Ising quantum chain

**Mario Collura**[1,2,3⋆]

**1** SISSA – International School for Advanced Studies, I-34136 Trieste, Italy.
**2** Theoretische Physik, Universität des Saarlandes, D-66123 Saarbrücken, Germany.
**3** Dipartimento di Fisica e Astronomia "G. Galilei",
Università di Padova, I-35131 Padova, Italy.

⋆ mario.collura@sissa.it

## Abstract

We study the out-of-equilibrium probability distribution function of the local order parameter in the transverse field Ising quantum chain. Starting from a fully polarised state, the relaxation of the ferromagnetic order is analysed: we obtain a full analytical description of the late-time stationary distribution by means of a remarkable relation to the partition function of a 3-states classical model. Accordingly, depending on the phase whereto the post-quench Hamiltonian belongs, the probability distribution may locally retain memories of the initial long-range order. When quenching deep in the broken-symmetry phase, we show that the stationary order-parameter statistics is indeed related to that of the ground state. We highlight this connection by inspecting the ground-state equilibrium properties, where we propose an effective description based on the block-diagonal approximation of the $n$-point spin correlation functions.



# 1 Introduction

Isolated many-body quantum systems at zero temperature may exhibit different phases depending on the value of a physical parameter appearing in the Hamiltonian. By varying this parameter, quantum fluctuations may drive the many-body ground state across a quantum phase transition [1]. This usually reflects in a change of the quantum order parameter identifying the transition. An example of such transition occurs in the one-dimensional quantum Ising model, where in the thermodynamic limit we can identify a broken-symmetry phase, with doubly degenerate ground state separated from the rest of the spectrum, and an unbroken-symmetry phase, with a unique ground state [2,3]. In the thermodynamic limit, the broken-symmetry ground-state exhibits long-range ferromagnetic order, whereas in the other phase the order is absent.

The out-of-equilibrium situation is much more complicated and much effort, both theoretically and experimentally, has been spent in recent years in order to have a better understanding of the non-equilibrium properties of isolated many-body quantum systems (see Refs. [4–7] as reviews on the subject, and their bibliography as a more comprehensive reference source). In particular, a crucial question is whether and how much of the order of the initial state is retained during the non-equilibrium dynamics and eventually up to the stationary state. We addressed this question by preparing the system in the ground state belonging to the ordered phase; thereafter we suddenly change the physical parameter that drives the phase transition and let the system evolve unitarily with the new Hamiltonian. As a consequence of this well know procedure (so-called quantum quench [8]), an extensive amount of energy is injected into the system and local observables are expected to relax toward stationary values. The local description of the stationary state will depend on the properties of the model under investigation: for non-integrable model the system does thermalise [9–21]; for integrable models a generalised thermalisation is expected [22–30].

Nevertheless, as far as we are interested in the time-dependent (and stationary) properties of the local order (characterising the order in a subsystem of typical length $\ell$), merely the order-

parameter average may not be sufficient to have a complete description of the order/disorder transition. As a consequence of the finite density of the excitations created by the quench, the expectation value of the local order parameter does relax to zero, no matter the quantum phase whereto the post-quench Hamiltonian belongs [27, 28, 31, 32].

However, in quantum mechanics, the measurement outcomes of a generic observable are described by a Probability Distribution Function (PDF) which encodes all information about quantum fluctuations of that observable in the system. Therefore, looking at the full PDF of the local order parameter will be a more effective way to understand how the initial order melts during the time evolution, or eventually survives until the steady state.

In the following, we apply this idea to the Ising quantum chain. Studying how the ferromagnetic properties of this model relax in time under the unitary dynamics is a very non-trivial question. In general, after a global quantum quench, we expect long-range ferromagnetic order disappears when inspecting a sufficiently large portion of the entire system. However, depending on the phase of matter the post-quench Hamiltonian belongs within, some remnants of the original order may locally survives. Local remnants of the original order have been recently observed by the author in a similar setup for the antiferromagnetic XXZ spin-1/2 chain [33].

In spite of a generous literature regarding PDFs of various observables (mainly conserved quantities, transverse magnetisation or work statistics in the Ising spin chain, particle number in the Bose(Fermi)-Hubbard model, etc.) in different quantum models, both non-interacting and interacting, either at the equilibrium [34–45] or out-of-equilibrium [46–53], the results on the order-parameter statistics are very few [33, 54, 55]. In particular, analytical findings on the PDF of the longitudinal magnetisation in the Ising quantum chain are only limited to some universal scaling behaviour in the ground state at the critical point [56].

Here we mainly concentrate our analysis on the out-of-equilibrium and long-time stationary properties of the full counting statistics of the order parameter in the Ising quantum chain after a quench from the fully polarised initial state. Remarkably, we are able to provide a closed analytical formula for the PDF in the stationary state by highlighting a striking connection with the partition function of a 3-states classical model. Moreover, we definitively stress a very appealing relation between the stationary PDF and the properties of the ground state of the post-quench Hamiltonian, in close analogy to what has been put forward in Ref. [33], undoubtedly confirming the great generality of such relationship which may be applied to other models. Indeed, it turns out that, when the initial ordered state is quenched into the unbroken symmetry phase, the local order quickly disappear and the PDF acquire a simple Gaussian shape which dynamically depends only on the first two cumulants of the local order parameter. Otherwise, when the state is quenched within the broken-symmetry phase, a simple Gaussian description is no more sufficient and the ground state of the post-quench Hamiltonian starts to play a crucial role in locally preserving the initial long-range order which eventually survives in the long-time limit.

The content of the manuscript is organised in the following way:

- Sec. 2 is devoted to introduce the model and its description in terms of diagonal spinless fermions; we introduce the Majorana fermions as well.

- In Sec. 3 we introduce the definition of the probability distribution function and its connection to the generating function of the moments (and cumulants as well) of the order parameter. We outline the general procedure to evaluate the full counting statistics of the longitudinal magnetisation in the spin-1/2 Ising quantum chain and write it in terms of a Pfaffian (or determinant) of a block-structured matrix which entries are given by the two-point correlation function of the Majorana operators.

- Sec. 4 collects the main results of our investigation, namely the non-equibrium dynamics

generated by unitary evolving a fully polarised initial state. After exploring the time-dependent behaviour, we mainly focus on the stationary properties. We show that in this case the generating function can be exactly evaluated by exploiting the block-diagonal form of the stationary $n$-point correlation functions, which can be further simplified to obtain a closed expression in terms of the partition function of a 3-states classical model.

- In Sec. 5 we analyse the ground state properties of the model: in particular, we focus on the small-field expansion as well as we exploit the block-diagonal approximation which turns out to give a good qualitative description of the ground state PDF as far as we keep the system sufficiently far from the critical point.

- In Sec. 6 we further propose an appealing interpretation of the stationary distribution in terms of the properties of the post-quench ground-state. We argue that such relation is exact in a special scaling regime deep in the ferromagnetic phase.

- Finally in Sec. 7 we draw our conclusions; we relegate all supplementary calculations in the Appendices A-D.

## 2 The model

We consider the one-dimensional spin-1/2 transverse field Ising chain whose Hamiltonian is

$$H = -\sum_{j=-\infty}^{\infty} \left( \sigma_j^x \sigma_{j+1}^x + h\,\sigma_j^z \right), \tag{1}$$

where $\sigma_j^\alpha$ are Pauli matrices acting on site $j$. The transverse field $h$ drives the ground state from a ferromagnetic region ($h < 1$) to a paramagnetic region ($h > 1$) across a quantum critical point, where the order parameter of the transition is the longitudinal magnetisation density $\mu \equiv \lim_{\ell \to \infty} \langle M_\ell \rangle / \ell$ (c.f. Sec. 3). The Ising Hamiltonian is invariant under the action of the string operator $P \equiv \prod_{j=-\infty}^{\infty} \sigma_j^z$. This $\mathbb{Z}_2$ symmetry is spontaneously broken in the ferromagnetic region: the two degenerate ground states $|GS_{\pm}\rangle$, corresponding to $P = \pm 1$, may recombine in such a way to exhibit non-vanishing values of the longitudinal magnetisation. In the following we make the choice to work in the invariant sector with $P = +1$.

The Hamiltonian (1) can be rewritten in terms of non-interacting spinless fermions via the Jordan-Wigner transformation [57]

$$\sigma_\ell^x = \prod_{j<\ell}(1-2n_j)(c_\ell^\dagger + c_\ell), \quad \sigma_\ell^y = i\prod_{j<\ell}(1-2n_j)(c_\ell^\dagger - c_\ell), \quad \sigma_\ell^z = 1-2n_\ell, \tag{2}$$

where $\{c_i, c_j^\dagger\} = \delta_{ij}$ and $n_i \equiv c_i^\dagger c_i$. Thus one has

$$H = -\sum_{j=-\infty}^{\infty} \left[ (c_j^\dagger - c_j)(c_{j+1}^\dagger + c_{j+1}) + h(1 - 2c_j^\dagger c_j) \right], \tag{3}$$

which can be easily diagonalised by a Bogoliubov transformation

$$c_j = \int_{-\pi}^{\pi} \frac{dk}{2\pi} e^{-ikj} \left[ \cos(\theta_k/2)\alpha_k + i\sin(\theta_k/2)\alpha_{-k}^\dagger \right], \tag{4}$$

where $\{\alpha_p, \alpha_q^\dagger\} = \delta_{pq}$ and Bogoliuobov angle

$$e^{i\theta_k} = \frac{h - e^{ik}}{\sqrt{1 + h^2 - 2h\cos(k)}}. \tag{5}$$

In terms of the diagonal fermions, the Hamiltonian becomes (apart from an overall normalisation factor)

$$H = \int_{-\pi}^{\pi} \frac{dk}{2\pi} \epsilon_k \left( \alpha_k^\dagger \alpha_k - \frac{1}{2} \right), \tag{6}$$

with dispersion relation $\epsilon_k = 2\sqrt{1 + h^2 - 2h\cos(k)}$. The ground state is the vacuum state of the Bogoliuobov fermions, namely $\alpha_k |GS_+\rangle = 0$, $\forall k$.

Within the approach we will be using in the next sections, it is convenient to replace the fermions $c_j$ with the Majorana fermions (here we can distinguish between two sets of operators through the apexes $x$ and $y$, or just introduce a doubled unique set with different operators corresponding to odd or even indices)

$$A_{2j-1} = a_j^x = (c_j^\dagger + c_j), \quad A_{2j} = a_j^y = i(c_j^\dagger - c_j), \tag{7}$$

which satisfy the algebra $\{a_i^x, a_j^x\} = \{a_i^y, a_j^y\} = 2\delta_{ij}$, $\{a_i^x, a_j^y\} = 0$, and such that one has

$$\sigma_j^x = \prod_{m<j} (i a_m^y a_m^x) a_j^x, \quad \sigma_j^y = \prod_{m<j} (i a_m^y a_m^x) a_j^y, \quad \sigma_j^z = i a_j^y a_j^x. \tag{8}$$

## 3 Probability Distribution Function of the order parameter

We are interested in the probability distribution function of the following observable

$$M_\ell = \frac{1}{2} \sum_{j=1}^{\ell} \sigma_j^x, \tag{9}$$

which describes the magnetisation along $\hat{x}$ of a subsystem consisting of sites $\{1, \dots, \ell\}$. The probability of such observable to take some value $m$ when the system is in a generic state is given by (in the following the bracket $\langle \cdots \rangle$ may represent either expectation value in a pure state or trace average over a density matrix)

$$P_\ell(m) = \langle \delta(M_\ell - m) \rangle = \int_{-\infty}^{\infty} \frac{d\lambda}{2\pi} e^{-im\lambda} F_\ell(\lambda), \tag{10}$$

where we defined the generating function of the moments of the probability distribution

$$F_\ell(\lambda) \equiv \langle e^{i\lambda M_\ell} \rangle, \quad \langle M_\ell^n \rangle = (-i)^n \partial_\lambda^n F_\ell(\lambda) \big|_{\lambda=0}, \tag{11}$$

which satisfies the following properties

$$F_\ell(0) = 1, \quad F_\ell(-\lambda) = F_\ell(\lambda)^*, \quad F_\ell(\lambda + 2\pi) = (-1)^\ell F_\ell(\lambda). \tag{12}$$

Thanks to (12), the probability distribution function in Eq. (10) can be expressed as

$$P_\ell(m) = \widetilde{P}_\ell(m) \times \begin{cases} \sum_{r \in \mathbb{Z}} \delta(m - r) & \text{for } \ell \text{ even,} \\ \\ \sum_{r \in \mathbb{Z}} \delta(m - 1/2 - r) & \text{for } \ell \text{ odd,} \end{cases} \tag{13}$$

which explicitly states that the PDF is different from zero only for integer (half-integer) values of $m$ when $\ell$ is even (odd). In particular, we defined the discrete Fourier transform

$$\widetilde{P}_\ell(m) \equiv \int_{-\pi}^{\pi} \frac{d\lambda}{2\pi} e^{-im\lambda} F_\ell(\lambda) = \int_{-\pi}^{\pi} \frac{d\lambda}{2\pi} \{\cos(m\lambda) \Re[F_\ell(\lambda)] + \sin(m\lambda) \Im[F_\ell(\lambda)]\}, \tag{14}$$

where the last passage is a consequence of $\widetilde{P}_\ell(m)$ being real and implies $\Re[F_\ell(-\lambda)] = \Re[F_\ell(\lambda)]$ and $\Im[F_\ell(-\lambda)] = -\Im[F_\ell(\lambda)]$. As expected from the spectrum of $M_\ell$, $\widetilde{P}_\ell(m)$ has to be evaluated only for $m \in \{-\ell/2, -\ell/2 + 1, \dots, \ell/2\}$, and satisfies the normalisation condition $\sum_{m=-\ell/2}^{\ell/2} \widetilde{P}_\ell(m) = 1$.

## 3.1 Asymptotic behaviour of the probability distribution

Before proceeding with the direct computation of the probability distribution function, let us summarise some properties of the PDF which are valid in the limit $\ell \gg 1$ whenever the state satisfies *cluster decomposition* and it is characterised by a *finite correlation length*.

For this purpose, it is useful to introduce the generating function of the cumulants $\langle M_\ell^n \rangle_c$ of the probability distribution

$$\log[F_\ell(\lambda)] \equiv \sum_{n=1}^{\infty} \langle M_\ell^n \rangle_c \frac{(i\lambda)^n}{n!}, \tag{15}$$

where the subscript $c$ stays for *connected* correlation function. From this definition, the cumulants are related to the moments by the following recursion formula

$$\langle M_\ell^n \rangle_c = \langle M_\ell^n \rangle - \sum_{m=1}^{n-1} \binom{n-1}{m-1} \langle M_\ell^{n-m} \rangle \langle M_\ell^m \rangle_c. \tag{16}$$

Interestingly, as far as the $n$-point connected correlation functions decay sufficiently fast so that $\lim_{\ell \to \infty} \sum_{j_1 < \cdots < j_n}^{\ell} \langle \sigma_{j_1}^x \cdots \sigma_{j_n}^x \rangle_c / \ell < \infty$, all cumulants turns out to be extensive quantities in the subsystem volume and admit the following asymptotic expansion in terms of their correspondent thermodynamic densities $\kappa_n \equiv \lim_{\ell \to \infty} \langle M_\ell^n \rangle_c / \ell$,

$$\langle M_\ell^n \rangle_c = \ell \kappa_n + o(\ell). \tag{17}$$

This is the case for the ground state in the paramagnetic region $|h| > 1$, where the expectation value of the order parameter vanishes and correlation functions decay exponentially; as well as after a global quantum quench, where the finite energy density injected into the system will build up a finite correlation length. Otherwise, in the ferromagnetic region, Eq. (17) does not apply for the $\mathbb{Z}_2$-symmetric ground-state $|GS_\pm\rangle$, since cluster decomposition is violated. Nevertheless, by considering the physical combination $|\mu_\pm\rangle = (|GS_+\rangle \pm |GS_-\rangle)/\sqrt{2}$, which is characterised by a finite value of the order parameter $\mu_\pm = \pm(1-h^2)^{1/8}/2$, the extensive behaviour of the cumulants is restored and the following arguments apply as well. In practice, the cumulant expansion (15) joined with the extensive property (17), leads to the following asymptotic behaviour of the PDF

$$\widetilde{P}_\ell(m) \simeq \int_{-\pi}^{\pi} \frac{d\lambda}{2\pi} e^{-im\lambda} e^{\ell \mathcal{F}(\lambda)}, \tag{18}$$

in terms of the large deviation function

$$\mathcal{F}(\lambda) \equiv \sum_{n=1}^{\infty} \kappa_n \frac{(i\lambda)^n}{n!} = \lim_{\ell \to \infty} \frac{\log[F_\ell(\lambda)]}{\ell}. \tag{19}$$

In the limit $\ell \to \infty$, the integral in Eq. (18) is dominated by the maximum of the large deviation function $\mathcal{F}(\lambda)$; whenever the expectation value of the order parameter $\mu$ is different from zero, we can keep the first two cumulants thus obtaining the following Gaussian

$$\widetilde{P}_\ell(m) \simeq \int_{-\pi}^{\pi} \frac{d\lambda}{2\pi} e^{-i(m-\ell\mu)\lambda} e^{-\ell\sigma^2\lambda^2/2} \simeq \frac{1}{\sqrt{2\pi\ell\sigma^2}} \exp\left[-\frac{(m-\ell\mu)^2}{2\ell\sigma^2}\right], \tag{20}$$

with standard deviation $\sigma = \sqrt{\kappa_2}$. Notice that, as far as $\mu$ and $\sigma$ are both different from zero, Eq. (20) does not admit an universal scaling behaviour of the variable $m$. This is essentially due to the different scaling with $\ell$ induced by the average (i.e. $m/\ell$) and by the standard deviation (i.e. $m/\sqrt{\ell}$).

## 3.2 Generalities of the order-parameter generating function

In general, the direct computation of the PDF of the longitudinal subsystem magnetisation is a very hard task, mainly due to the nonlocal nature of the $\sigma^x$ operator in terms of Majorana fermions. For this reason, the brute force approach to compute the order parameter PDF goes through the evaluation of the generating function $F_\ell(\lambda)$. Using the following identity

$$\exp\{i(\lambda/2)\sigma_j^x\} = \cos(\lambda/2) + i\sin(\lambda/2)\sigma_j^x, \tag{21}$$

which is a direct consequence of the Pauli matrices property $(\sigma_j^x)^2 = 1$, Eq. (11) can be rewritten as

$$F_\ell(\lambda) = \Big\langle \prod_{j=1}^{\ell}\Big[\cos(\lambda/2) + i\sin(\lambda/2)\sigma_j^x\Big]\Big\rangle. \tag{22}$$

We can rearrange such formula, counting the number of Pauli matrices appearing in the string. Moreover, since we are working in one of the two $\mathbb{Z}_2$-invariant sub-sectors (with $P = +1$), the expectation value of an odd number of $\sigma^x$ operators is vanishing, thus in general on has

$$F_\ell(\lambda) = \cos(\lambda/2)^\ell \sum_{n=0}^{\lfloor \ell/2\rfloor}[i\tan(\lambda/2)]^{2n} \sum_{j_1<j_2<\cdots<j_{2n}}^{\ell}\langle\sigma_{j_1}^x\sigma_{j_2}^x\cdots\sigma_{j_{2n}}^x\rangle, \tag{23}$$

where the ordered indexes $\{j_1,\ldots,j_{2n}\}$ are in the interval $[1,\ell]$. The previous equation implies $\Im[F_\ell(\lambda)] = 0$ and $\widetilde{P}_\ell(-m) = \widetilde{P}_\ell(m)$, and can be used in the definition (14) to get the following representation for the order parameter PDF

$$\widetilde{P}_\ell(m) = \sum_{n=0}^{\lfloor \ell/2\rfloor} p_{m,n}(\ell) \sum_{j_1<j_2<\cdots<j_{2n}}^{\ell}\langle\sigma_{j_1}^x\sigma_{j_2}^x\cdots\sigma_{j_{2n}}^x\rangle, \tag{24}$$

where $p_{m,n}(\ell)$ are defined in the Appendix A. The evaluation of $F_\ell(\lambda)$ and $\widetilde{P}_\ell(m)$ reduces therefore to the computation of the generic string $\langle\sigma_{j_1}^x\sigma_{j_2}^x\cdots\sigma_{j_{2n}}^x\rangle$. Notice that the PDFs provide all possible informations related to their corresponding observables; for example, as a side result, from Eq. (24) the Emptiness Formation Probability (EFP) $\mathcal{E}_\ell$, namely the probability to have $\ell$ contiguous spins up, is straightforwardly obtained from $\widetilde{P}_\ell(\ell/2)$, where it is easy to show that $p_{\ell/2,n}(\ell) = 2^{-\ell}$ as expected, and therefore $\widetilde{P}_\ell(\ell/2) = 2^{-\ell}\langle\prod_{j=1}^{\ell}(1+\sigma_j^x)\rangle \equiv \mathcal{E}_\ell$. The EFP can be also obtained directly form the generating function by analytical continuation in the complex plane, namely

$$\mathcal{E}_\ell = \lim_{\lambda\to\infty}\frac{F_\ell(-i\lambda)}{[2\cosh(\lambda/2)]^\ell}. \tag{25}$$

As a final remark, an interesting consequence of Eq. (25), when joined with the asymptotic expansion of the generating function, regards the behaviour of the extensive part of the logarithm of the EFP, i.e. $\ell^{-1}\log(\mathcal{E}_\ell) \sim \lim_{\lambda\to\infty}[\mathcal{F}(-i\lambda) - \lambda/2]$ which, in order to be finite, implies that the analytic continuation in the imaginary axis of the large deviation function has to asymptotically behave like $\mathcal{F}(-i\lambda) \sim \lambda/2 + cst$, for $\lambda\to\infty$. Notice that, this result cannot be recovered from the expansion (15) when truncated to any finite order, since it is an asymptotic property which requires either the full knowledge of the large deviation function $\mathcal{F}(\lambda)$, or its expansion nearby $\lambda = -i\infty$.

### 3.2.1 Strings of $\sigma^x$ in the Ising quantum chain

The evaluation of the expectation value of a generic string of $\sigma^x$ operators can be carried out by using the Majorana fermions. The expectation value $\langle\sigma_{j_1}^x\sigma_{j_2}^x\cdots\sigma_{j_{2n}}^x\rangle$ can be rewritten in

such a way that strings of Majorana operators appear only in the intervals $[j_1, j_2]$, $[j_3, j_4]$, up to $[j_{2n-1}, j_{2n}]$. In particular, for a generic interval $[p, q]$ with $p < q$, one has

$$\sigma_p^x \sigma_q^x = a_p^x \prod_{k=p}^{q-1} (i a_k^y a_k^x) a_q^x = \prod_{k=p}^{q-1} (-i a_k^y a_{k+1}^x) = (-i)^{q-p} \prod_{k=p}^{q-1} A_{2k} A_{2k+1} = (-i)^{q-p} \prod_{k=2p}^{2q-1} A_k,$$

which easily leads to

$$\langle \sigma_{j_1}^x \sigma_{j_2}^x \cdots \sigma_{j_{2n}}^x \rangle = (-i)^{\mathcal{L}_{j_n}} \langle \prod_{k \in \mathcal{U}} A_k \rangle, \tag{26}$$

where $\mathcal{L}_{j_n} = \sum_{k=1}^{n} (j_{2k} - j_{2k-1})$, $\mathcal{U}_{j_n} = [2j_1, 2j_2 - 1] \cup [2j_3, 2j_4 - 1] \cup \cdots \cup [2j_{2n-1}, 2j_{2n} - 1]$, and the string contains an even number $2\mathcal{L}_{j_n}$ of Majorana operators. Here we used the shorthand notation $\boldsymbol{j}_n \equiv \{j_1, \ldots, j_{2n}\}$ for the full set of indices. The expectation value of a generic string of Majorana operators involves the evaluation of the Pfaffian of a skew-symmetric real matrix $\mathbb{M}_{j_n}$ which explicitly depend on the particular choice of the indices. This matrix has dimension $2\mathcal{L}_{j_n} \times 2\mathcal{L}_{j_n}$, and entries given by

$$(\mathbb{M}_{j_n})_{m_p, n_q} = -i \langle A_p A_q \rangle + i \delta_{p,q}, \quad \text{for} \quad \{p, q\} \in \mathcal{U}_{j_n}, \tag{27}$$

where the indices $m_p$ and $n_q$ move in $\{0, \ldots, 2\mathcal{L}_{j_n} - 1\}$, and have the function of shrinking all together the intervals appearing in $\mathcal{U}_{j_n}$. In terms of this matrix, the expectation value of an even number of $\sigma^x$ operators is finally given by [58, 59]

$$\langle \sigma_{j_1}^x \sigma_{j_2}^x \cdots \sigma_{j_{2n}}^x \rangle = \text{pf}(\mathbb{M}_{j_n}) = (-1)^{\mathcal{L}_{j_n}(\mathcal{L}_{j_n} - 1)/2} \text{pf} \begin{bmatrix} -\mathbb{F}_{j_n} & \mathbb{G}_{j_n} \\ -\mathbb{G}_{j_n}^T & \mathbb{F}_{j_n} \end{bmatrix}, \tag{28}$$

where $\text{pf}(\cdots)$ denote the Pfaffian[1]. In the last passage we performed $\mathcal{L}_{j_n}(\mathcal{L}_{j_n} - 1)/2$ column and row permutations and exploited both translational invariance and reflection symmetry of the state. As a consequence, the real matrices $\mathbb{F}_{j_n}$ and $\mathbb{G}_{j_n}$ ($\mathbb{F}_{j_n}$ being also skew-symmetric) have dimensions $\mathcal{L}_{j_n} \times \mathcal{L}_{j_n}$ and entries given by [27]

$$(\mathbb{F}_{j_n})_{m_p, n_q} = i \langle a_p^y a_q^y \rangle - i \delta_{p,q} = -i \langle a_p^x a_q^x \rangle - i \delta_{p,q} \equiv f_{p-q}, \tag{29}$$

$$(\mathbb{G}_{j_n})_{m_p, n_q} = -i \langle a_p^y a_{q+1}^x \rangle \equiv g_{p-q}, \tag{30}$$

with $\{p, q\} \in [j_1, j_2 - 1] \cup [j_3, j_4 - 1] \cup \cdots \cup [j_{2n-1}, j_{2n} - 1]$ and where the indices $m_p$ and $n_q$ now run in $\{0, \ldots, \mathcal{L}_{j_n} - 1\}$, and once again have the function of shrinking all together the intervals. The knowledge of the fermionic correlation functions (29) and (30) together with the representation (28) are the basic ingredients to compute the generating function in Eq. (23). However, without any further simplification this computation would require the evaluation of a number of Pfaffians which growths exponentially with the subsystem size $\ell$, making this approach ineffective already for $\ell \gtrsim 20$.

In some cases, like at the equilibrium (c.f. Sec. 5) or in the stationary state after a quench (c.f. Sec. 4), we have $\mathbb{F}_{j_n} = \mathbb{O}$ and, exploiting the properties of the Pfaffian, the full counting statistics can be written as

$$F_\ell(\lambda) = \cos(\lambda/2)^\ell \sum_{n=0}^{\lfloor \ell/2 \rfloor} [i \tan(\lambda/2)]^{2n} \sum_{j_1 < j_2 < \cdots < j_{2n}} \det(\mathbb{G}_{j_n}). \tag{31}$$

This expression is still very difficult to handle, therefore in the following we will try to simplify Eq. (31) in order to get some analytical approximations (or exact result) which will be eventually compared to the numerical data obtained by using the infinite time-evolving block-decimation (iTEBD) algorithm [60].

---

[1] Interestingly, when keeping the odd/even checkerboard structure of the matrix $\mathbb{M}_{j_n}$ in Eq. (27), the sign of the Pfaffian is such that one eventually has $\text{pf}(\mathbb{M}_{j_n}) = \det(\mathbb{M}_{j_n})^{1/2} = (-1)^{\mathcal{L}_{j_n}/2} \det(\mathbb{G}_{j_n})^{1/2} \det(\mathbb{F}_{j_n} \mathbb{G}_{j_n}^{-1} \mathbb{F}_{j_n} - \mathbb{G}_{j_n}^T)^{1/2}$, when $\det(\mathbb{G}_{j_n}) \neq 0$.

# 4 Quench dynamics from the full ordered state

We are interested in how the long-range order which characterises the ferromagnetic phase melts in time after a global quantum quench. Hence we consider the most representative quench so as to explore the relaxation of the full probability distribution function of the order parameter in the Ising quantum chain. We prepare the system in the fully polarised $\mathbb{Z}_2$-symmetric ground state, namely the ground state of the Ising Hamiltonian at $h = 0$ (within the symmetry sector $P = +1$),

$$|\Psi_0\rangle = \frac{1}{\sqrt{2}}(|\cdots \uparrow\uparrow\uparrow\uparrow \cdots\rangle + |\cdots \downarrow\downarrow\downarrow\downarrow \cdots\rangle), \tag{32}$$

which is characterised by the simple generating function $F_\ell^{(0)}(\lambda) = \cos(\ell\lambda/2)$ and related PDF $\widetilde{P}_\ell^{(0)}(m) = (\delta_{m,-\ell/2} + \delta_{m,\ell/2})/2$. The post-quench dynamics of the order-parameter generating function is completely determined by the following Fermionic two-point functions [27, 28]

$$
\begin{aligned}
f_n &= i \int_{-\pi}^{\pi} \frac{dk}{2\pi} e^{ikn} \frac{h\sin(k)}{\epsilon_k/2} \sin(2\epsilon_k t), \\
g_n &= -\int_{-\pi}^{\pi} \frac{dk}{2\pi} e^{ik(n-1)} e^{i\theta_k} \left[ \frac{1-h\cos(k)}{\epsilon_k/2} - i\frac{h\sin(k)}{\epsilon_k/2} \cos(2\epsilon_k t) \right].
\end{aligned}
\tag{33}
$$

We focus on this particular quench since we may expect that, whenever the dynamics starts from deep in the ferromagnetic region, there should not be qualitative differences in the behaviour of the PDF, even retaining a finite small value of the initial transverse field, thus inducing finite fluctuations in the initial magnetisation. However, thanks to the simple structure of the state $|\Psi_0\rangle$, we are able to obtain full analytical results for the stationary distribution after this quench.

Due to the non vanishing matrix elements $f_n$, we have to direct evaluate Eq. (28) in order to obtain prediction for the time-dependent probability distribution function of the subsystem longitudinal magnetisation; luckily, even for relatively small subsystem sizes, the dynamics of the PDF already shows very peculiar features. In Figure 1 we report the density plot of the time-dependent PDF for a subsystem with size $\ell = 20$ and different values of the post-quench transverse field, both in the ferromagnetic and in the paramagnetic region.

On closer inspection of the data, it seems clear that, when quenching into the paramagnetic region, the initial ferromagnetic order suddenly melts, and the PDF gets similar to a normal distribution centred in zero. How fast the relaxation occurs definitively depends on how big the value of the post-quench field is. Otherwise, when the unitary dynamics is governed by a Hamiltonian with value of the transverse field $0 < h < 1$, then the probability distribution seems to retain a much broader shape, which eventually disappears only for sufficiently large subsystem sizes; how large depending on the actual value of the post-quench field.

Interestingly, deep in the ferromagnetic regime, the relaxation dynamics of the probability distribution seems very peculiar: the two peaks of the initial distribution (located at $m = \pm\ell/2$ with value $1/2$) partially reduce in amplitude by emitting each a ballistic-propagating probability-stream with velocity $\sim 2|\partial_k \epsilon_k|_{\max} = 4|h|$. These counter-propagating streams are very faint when quenching deep in the ferromagnetic region, mainly because the density of excitations generated by the quench are very low as $h$ is smaller (notice that when $h \to 0$, there is no quench dynamics). Nevertheless, for any small but finite value of $h$, the dynamics is not trivial, and the equilibration toward the stationary state occurs by means of a de-phasing mechanism involved in the probability-flow propagation.

In the following we exactly compute the full counting statistics of the order-parameter in the stationary states for both phases.

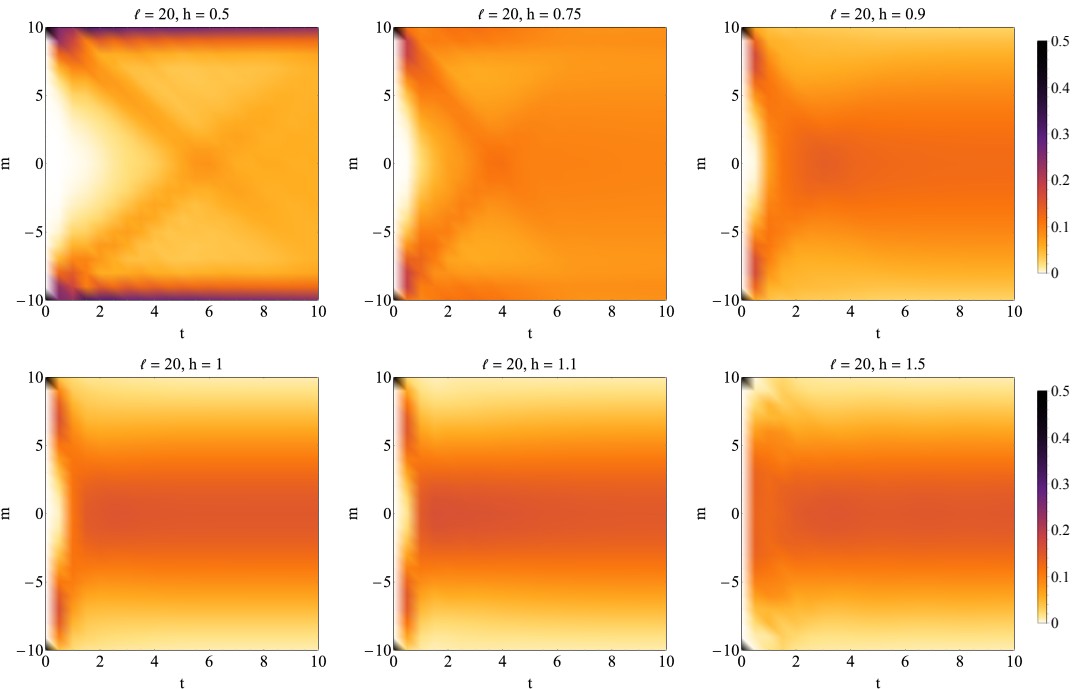

Figure 1: Color density plot of the time evolution of the PDF of the subsystem order parameter for subsystem sizes $\ell = 20$ and different values of the post-quench transverse field $h$.

## 4.1 Stationary probability distribution function

In the long time limit after the quench the correlation functions of Majorana operators in Eq. (33) simplify to

$$f_n = 0, \quad g_n = -\int_{-\pi}^{\pi} \frac{dk}{2\pi} e^{ik(n-1)} e^{i\theta_k} \frac{1 - h\cos(k)}{\epsilon_k/2}, \tag{34}$$

and the full counting statistics can be evaluated using Eq. (31). In this case, the Fermionic correlation function can be explicitly worked out, both in the ferromagnetic phase ($|h| < 1$) and in the paramagnetic regime ($|h| \geq 1$). By expanding the Fourier coefficient $e^{i\theta_k} \frac{1 - h\cos(k)}{\epsilon_k/2}$ in Eq. (34) for $h \geq 1$ one easily obtains

$$
\begin{aligned}
g_n &= \frac{\sin(\pi n)}{\pi} \left[ \frac{n-1}{n(n-2)} + \sum_{k=1}^{\infty} (-1)^k \frac{h^{-k}}{(n-k)(n-k-2)} \right] \\
&= \frac{1}{2}\delta_{n,0} - \frac{1}{2h}\delta_{n,1} + \frac{1}{2}\sum_{k=2}^{\infty}(h^{-k+2} - h^{-k})\delta_{n,k},
\end{aligned} \tag{35}
$$

similarly, in the opposite regime, i.e. for $0 \leq h \leq 1$, one hase

$$
\begin{aligned}
g_n &= \frac{\sin(\pi n)}{\pi} \left[ \frac{1}{n} - \sum_{k=1}^{\infty} (-1)^k \frac{h^k}{(n+k)(n+k-2)} \right] \\
&= -\frac{h}{2}\delta_{n,1} + \left(1 - \frac{h^2}{2}\right)\delta_{n,0} + \frac{1}{2}\sum_{k=1}^{\infty}(h^k - h^{k+2})\delta_{n,-k},
\end{aligned} \tag{36}
$$

where the first line in Eq.s (35) and (36) should be understood as the limit of $n$ that becomes integer, thus being non-zero whenever $\sin(\pi n)$ gets cancelled by the poles in the series within the square brackets.

Even though the matrix elements $g_n$ share many similarities between the two phases, small changes are sufficient to have big effects in the structure of the full matrix $\mathbb{G}_{j_n}$, thus having huge consequences on the order parameter probability distribution function in the stationary state. In both cases, the structure of the Majorana correlation functions guarantee that the matrix $\mathbb{G}_{j_n}$ in the stationary state after this quench *exactly* reduces to a block triangular form. Its determinant is given by the product of the determinants of the diagonal blocks, and thus we have

$$F_\ell(\lambda) = \cos(\lambda/2)^\ell \sum_{n=0}^{\lfloor \ell/2 \rfloor} [i \tan(\lambda/2)]^{2n} \sum_{j_1 < j_2 < \cdots < j_{2n}}^{\ell} \mathcal{D}_{j_2-j_1} \mathcal{D}_{j_4-j_3} \cdots \mathcal{D}_{j_{2n}-j_{2n-1}}, \qquad (37)$$

where $\mathcal{D}_z \equiv \det(\mathbb{G}_{[1,z],[1,z]}) = \langle \sigma_1^x \sigma_{1+z}^x \rangle$ (*c.f.* Sec 5.3) can be explicitly evaluate in both phases and leads to a closed expression for the stationary generating function. As a matter of fact, this is the first case where a full analytical description of the stationary properties of the order-parameter statistics in the Ising quantum chain after a quench has been obtained.

### 4.1.1 Paramagnetic phase

For $h \geq 1$, the matrix $\mathbb{G}_{[1,z],[1,z]}$ is lower triangular, therefore

$$\mathcal{D}_z = g_0^z = \left(\frac{1}{2}\right)^z, \qquad (38)$$

which turns out to be independent of $h$. The generating function reduces to

$$F_\ell(\lambda) = \cos(\lambda/2)^\ell \sum_{n=0}^{\lfloor \ell/2 \rfloor} [i \tan(\lambda/2)]^{2n} \sum_{j_1 < j_2 < \cdots < j_{2n}}^{\ell} \left(\frac{1}{2}\right)^{\mathcal{L}_{j_n}}, \qquad (39)$$

which is related to the partition function of a 1D Ising model of length $\ell+1$ and fixed boundary conditions (see Appendix C), namely

$$F_\ell(\lambda) = \cos(\lambda/2)^\ell e^{-\Lambda \ell} e^{A(\ell+1)} \mathcal{Z}_I(\Lambda, A, \ell+1), \qquad (40)$$

with $\Lambda = -\log[i \tan(\lambda/2)]/2$ and $A = -\log(2)/2$. Explicitly, one has

$$F_\ell(\lambda) = \frac{\cos(\lambda/2)^\ell}{2^{\ell/2}} \left[ \frac{\tan(\lambda/2)^2 \tilde{z}_1^\ell}{\tan(\lambda/2)^2 - (\tilde{z}_1 - \sqrt{2})^2} + \frac{\tan(\lambda/2)^2 \tilde{z}_2^\ell}{\tan(\lambda/2)^2 - (\tilde{z}_2 - \sqrt{2})^2} \right], \qquad (41)$$

where $\tilde{z}_j$ are related to the transfer matrix eigenvalues $z_j$ reported in Appendix C via $\tilde{z}_j = e^{-\Lambda} z_j$, and they are given by

$$\tilde{z}_{1,2} = \frac{3}{2\sqrt{2}} \pm \sqrt{\frac{1}{8} - \tan(\lambda/2)^2}. \qquad (42)$$

For $\ell \gg 1$ the behaviour of the full counting statistics in Eq. (41) is dominated by the eigenvalue with the largest modulus, i.e. $\tilde{z}_1$, thus obtaining the following analytical closed expression for the large deviation function

$$\mathcal{F}(\lambda) = \log[\cos(\lambda/2) \tilde{z}_1 / \sqrt{2}]. \qquad (43)$$

This result implies that the distribution of rare outcomes, which are determined by the tails of the stationary PDF, should exhibit a non-gaussian shape.

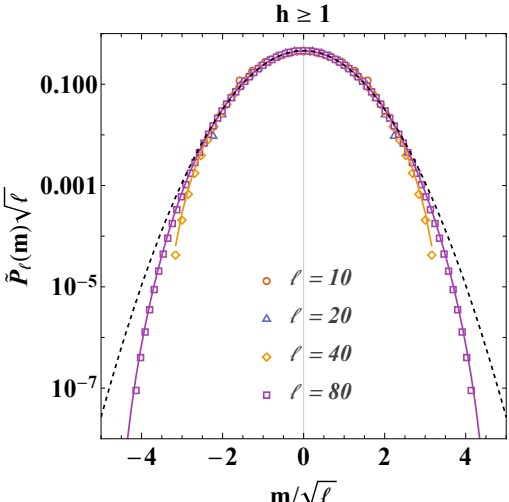

Figure 2: The rescaled PDF of the subsystem order parameter in the stationary state after quenching the transverse field into the paramagnetic region ($h \geq 1$.) Symbols are the exact results obtained from the exact generating function in Eq. (41); these are compared with the large deviation scaling obtained from Eq. (43) (full lines). The black-dashed line is the Gaussian approximation (44) valid in the thermodynamics limit.

However, when $\ell \to \infty$ and we are interested only on the most probable outcomes, the PDF can be evaluated by keeping only the first two cumulants and it is very well described by the following Gaussian distribution

$$\widetilde{P}_\ell(m) \simeq \frac{1}{\sqrt{2\pi\ell\sigma^2}} \exp\left[-\frac{m^2}{2\ell\sigma^2}\right], \tag{44}$$

with $\sigma^2 = 3/4$. As a side result, the emptiness formation probability in the stationary state after quenching into the paramagnetic region reads

$$\mathcal{E}_\ell = \frac{2}{3}\left(\frac{3}{4}\right)^\ell. \tag{45}$$

In Fig. 2 we compare the exact PDF obtained by taking the discrete Fourier transform of Eq. (41) with both the large deviation scaling from Eq. (43) as well as the thermodynamic Gaussian approximation. As expected, as far as $\ell \gtrsim 20$, data are well captured by the large deviation function which still retains a sub-leading $\ell$ dependence in the thermodynamic scaling regime and exhibits tails that differ from the Gaussian behaviour. Gaussianity is recovered only in the thermodynamic limit, and starting from a neighbourhood of the average $m/\sqrt{\ell} = 0$.

### 4.1.2 Ferromagnetic phase

For $0 \leq h < 1$, the matrix $\mathbb{G}_{[1,z],[1,z]}$ reduces to a Toeplitz-Hessenberg matrix and the determinant can be analytically evaluated (see Appendix B)

$$\mathcal{D}_z = \alpha^{z+1} + \beta^{z+1}, \quad \alpha = \frac{1+\sqrt{1-h^2}}{2}, \quad \beta = 1 - \alpha. \tag{46}$$

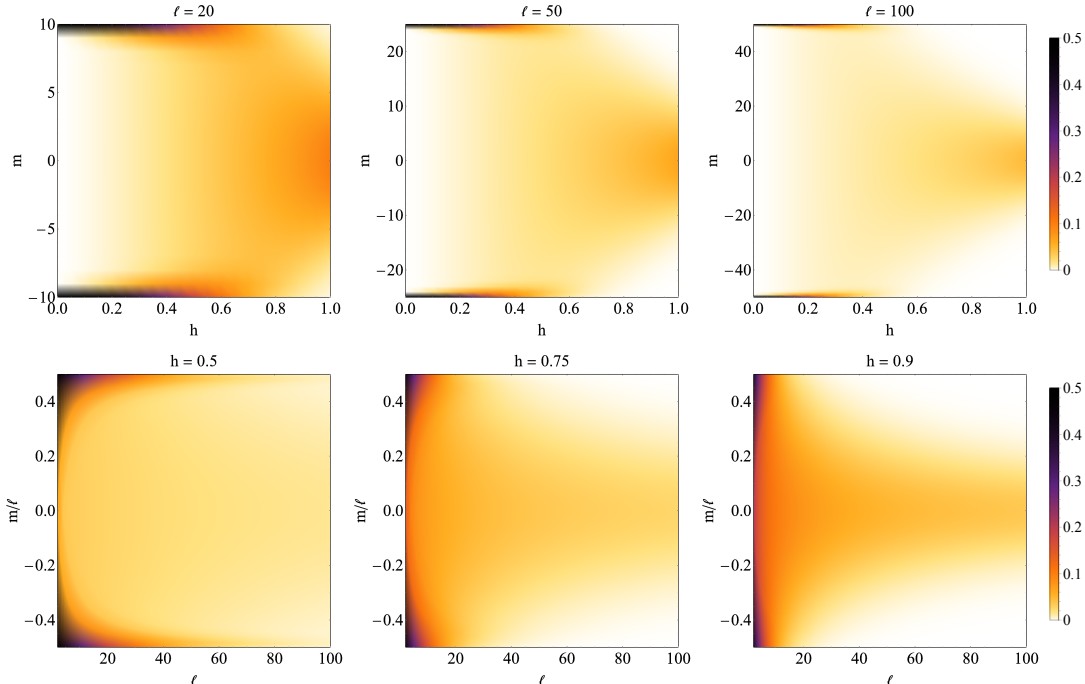

Figure 3: Color density plot of the PDF of the subsystem order parameter in the stationary state after quenching the transverse field $h$ within the ferromagnetic region. (**top**) Stationary PDF for different subsystem dimensions $\ell = 20, 50, 100$ as a function of the post-quench transverse field $h \in [0, 1]$. (**bottom**) The same PDF as a function of the subsystem size $\ell \in [2, 100]$ for fixed value of the post-quench field $h = 0.5, 0.75, 0.9$.

The generating function explicitly reads

$$F_\ell(\lambda) = \cos(\lambda/2)^\ell \sum_{n=0}^{\lfloor \ell/2 \rfloor} [i \tan(\lambda/2)]^{2n} \sum_{j_1 < j_2 < \cdots < j_{2n}} \prod_{i=1}^n \left( \alpha^{j_{2i} - j_{2i-1} + 1} + \beta^{j_{2i} - j_{2i-1} + 1} \right), \qquad (47)$$

which can be interpreted as the partition function of a 3-states classical chain with $\ell + 1$ sites and fixed boundary conditions, namely

$$F_\ell(\lambda) = \cos(\lambda/2)^\ell \mathcal{Z}_P(\Lambda, A, B, \ell + 1), \qquad (48)$$

with $\Lambda = \log[i \tan(\lambda/2)]$, $A = \log(\alpha)$ and $B = \log(\beta)$. This partition function can be easily evaluated as $\mathcal{Z}_P(\Lambda, A, B, \ell + 1) = \sum_{j=1}^3 \langle \emptyset | z_j \rangle \langle z_j | \emptyset \rangle z_j^\ell$, in terms of the eigenvalues $z_j$ of the Transfer Matrix (see Appendix D for details and definitions). Finally by Fourier transforming Eq. (48), the stationary probability distribution is obtained.

In Figure 3 we show the behaviour of the PDF in the stationary state, either at fixed subsystem size $\ell$ and varying $h \in [0, 1]$, or for $\ell \in [2, 100]$ at fixed $h$. Obviously, when $h$ approach the critical value, the shape of the stationary PDF approach what has been found in the previous section, namely the Fourier transform of Eq. (41); moreover, as expected, for any value of the magnetic field, and sufficiently large subsystem sizes (larger than a typical length $\ell^*(h)$ which depends on the actual value of the field), the distribution is expected to reduce to a Gaussian.

In fact, also in this regime, when $\ell \gg \ell^*(h)$ the probability distribution function is dominated by the *largest* eigenvalue, namely $z_3$ in this case, thus leading to large deviation scaling $\mathcal{F}(\lambda) = \log[\cos(\lambda/2)z_3]$. Notice that this is valid only for "sufficiently" large subsystem sizes

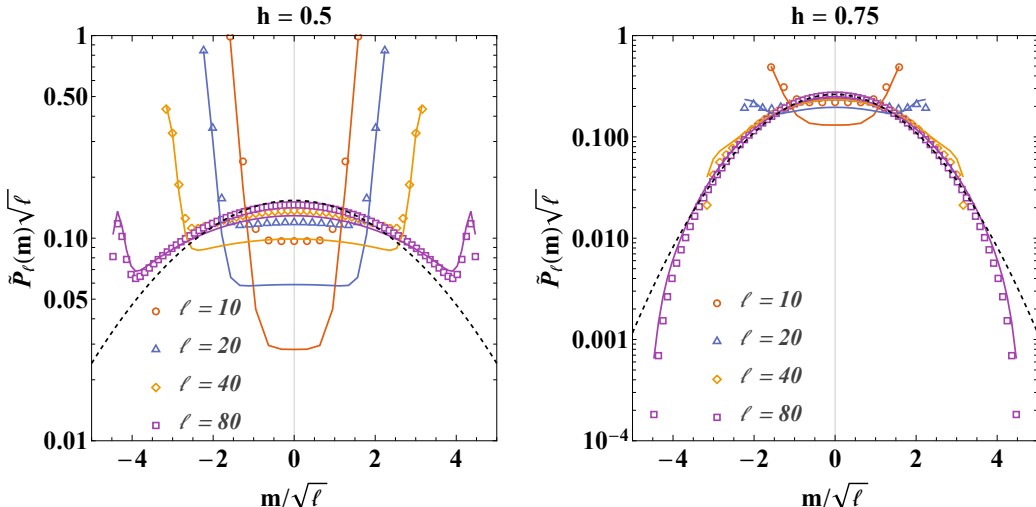

Figure 4: The rescaled PDF of the subsystem order parameter in the stationary state after quenching the transverse field into the ferromegnetic region (here $h = 0.5$ and $0.75$). Symbols are the exact results obtained from the exact generating function in Eq. (48); these are compared with the large deviation scaling obtained from $F_\ell(\lambda) \sim [\cos(\lambda/2)z_3]^\ell$ (full lines), where $z_3$ is defined in the Appendix D. The black-dashed line is the Gaussian approximation $F_\ell(\lambda) \sim \exp[-\ell(-5/4 + 2/h^2)\lambda^2/2]$ valid in the thermodynamics limit.

since, for any intermediate regime, contribution from $z_2$ may be relevant (see Appendix D). Furthermore, in the thermodynamic limit, the asymptotic behaviour of the generating function is dominated by the vicinity of $\lambda = 0$, and reads $\log[F_\ell(\lambda)] \sim -\ell(-5/4 + 2/h^2)\lambda^2/2$, thus leading to the well-known thermodynamic Gaussian rescaling.

Notwithstanding, it is clear from the plots that, for any finite $\ell$, there is always a region $h < h^*(\ell)$ (or $\ell < \ell^*(h)$) wherein the stationary distribution exhibits a strong bimodal shape, thus highlighting the presence of local ferromagnetic order, and large deviation scaling does not apply yet. This is specially evident in Fig. 4 where we compare the exact PDF with respect to its scaling behaviour for two different values of the transverse field. We can safely say that, at the level of any finite subsystem, the long-range order encoded in the initial state is somehow preserved in the stationary state.

Remarkably, using the analytical continuation of the partition function of the 3-states model into Eq. (25), the limit $\lambda \to \infty$ can be easily taken, thus giving an exact simple expression for the emptiness formation probability in the stationary state

$$\mathcal{E}_\ell = \left(\frac{1}{3} + \frac{1}{3\sqrt{4 - 3h^2}}\right)\left(\frac{1}{2} + \frac{\sqrt{4 - 3h^2}}{4}\right)^\ell + \left(\frac{1}{3} - \frac{1}{3\sqrt{4 - 3h^2}}\right)\left(\frac{1}{2} - \frac{\sqrt{4 - 3h^2}}{4}\right)^\ell, \quad (49)$$

which, as expected, reduces to Eq. (45) when $h \to 1$. Moreover, in the scaling limit $\ell \to \infty$ with constant $\mathfrak{h} = h\sqrt{\ell}$, it is worth noting that the emptiness formation probability reveals the scaling behaviour $\mathcal{E}_\ell \sim \exp(-3\mathfrak{h}^2/16)/2$.

## 5 Ground state properties

Here we explore the properties of the generating function of the subsystem longitudinal magnetisation and the associated probability distribution function in the ground state of the Ising

quantum chain. This provides as a preparation for us to be able to make a remarkable connection between ground-state properties and the stationary results found in the previous section (*c.f.* Sec. 6).

At zero temperature, the correlation functions of Majorana operators are given by [27,28]

$$f_n = 0, \quad g_n = -\int_{-\pi}^{\pi} \frac{dk}{2\pi} e^{ik(n-1)} e^{i\theta_k}, \tag{50}$$

and therefore Eq. (31) applies. In order to simplify this expression, let us start form the fact that in the vicinity of $h = 0$, $\mathbb{G}_{j_n}$ is very close to be a diagonal matrix. This can be easily seen from the expansion of $e^{i\theta_k}$ in power of $h$ which leads to

$$
\begin{aligned}
g_n &= \frac{\sin(\pi n)}{\pi}\left[\frac{1}{n} - \sum_{k=1}^{\infty}(-1)^k h^k \frac{\Gamma((n-k)/2)\Gamma((n+k-1)/2)}{4\,\Gamma((n-k+1)/2)\Gamma((n+k+2)/2)}\right] \\
&= \left(1 - \frac{h^2}{4}\right)\delta_{n,0} + \frac{h}{2}(\delta_{n,-1} - \delta_{n,1}) + \frac{h^2}{8}(3\delta_{n,-2} - \delta_{n,2}) + O(h^3),
\end{aligned}
\tag{51}
$$

where the same comment after Eq. (36) applies here. Previous expansion is formally valid for $|h| < 1$ and tell us that the first non-vanishing contribution to the fermionic correlation function at distance $|n|$ is order $h^{|n|}$. Interestingly, for $n = 0, 1$ the sum can be easily rewritten as $g_0 = 2\Re[E(h)]/\pi$, $g_1 = -2\Re[E(1/h)]/\pi$ in terms of the Elliptic integral $E(h) = \int_0^{\pi/2} d\theta\sqrt{1 - h^2\sin(\theta)^2}$, where taking the real part makes the result valid also for $|h| > 1$.

## 5.1 Small $h$ expansion

Deep in the ferromagnetic phase, we may therefore tray to expand the generating function as follow

$$F_\ell(\lambda) = \sum_{k=0}^{\infty} F_\ell^{(2k)}(\lambda) h^{2k}, \tag{52}$$

where for symmetry reason only the even terms appear in the sum and the order zero term trivially coincides with the generating function evaluated in the ground state at zero magnetic field, i.e.

$$F_\ell^{(0)}(\lambda) = \cos(\ell\lambda/2). \tag{53}$$

Using Eq. (51), by inspecting the structure of the determinant (after a bit of combinatorics), we obtain

$$\det(\mathbb{G}_{j_n}) = 1 - \frac{h^2}{4}n + \frac{h^4}{64}[2(n-2)^2 + 3s - 8] + O(h^6). \tag{54}$$

Notice that, the second order term depends only on $n$, i.e. the total number of $\sigma^x$ operators entering in the string. Otherwise, the forth order term depends also on $s \in [0, \ldots, 2n-1]$ which counts how many insertions in the ordered set $\boldsymbol{j}_n$ are such that $j_{i+1} = j_i + 1$: in other words it counts how many neighbouring $\sigma^x$ operators appear in the string, thus it depends on the particular choice of the set $\boldsymbol{j}_n$, namely $s = \sum_{i=1}^{2n-1} \delta_{j_{i+1}, j_i+1}$. In particular, by using

$$\sum_{j_1 < j_2 < \cdots < j_{2n}} 1 = \binom{\ell}{2n}, \quad \sum_{j_1 < j_2 < \cdots < j_{2n}} s = \sum_{i=1}^{2n-1}\sum_{j_1 < j_2 < \cdots < j_{2n}} \delta_{j_{i+1}, j_i+1} = (2n-1)\binom{\ell-1}{2n-1}, \tag{55}$$

where the last equivalence is due to the fact that the sum over $\{j_1, \ldots j_{2n}\}$ is independent on $i$, we easily obtain

$$
\begin{aligned}
F_\ell^{(2)}(\lambda) &= \frac{\sin(\lambda/2)}{8} \ell \sin[(\ell-1)\lambda/2], \quad (56) \\
F_\ell^{(4)}(\lambda) &= \frac{\sin(\lambda/2)}{256} \{[6 - \ell(\ell-9)]\sin[(\ell-1)\lambda/2] + (\ell-1)(\ell+6)\sin[(\ell-3)\lambda/2]\}.
\end{aligned}
$$

Unfortunately, although Eq. (52) seems very appealing, it is only an asymptotic series which is not convergent for arbitrary $\ell$. By a more careful inspection indeed, each term $F_\ell^{(2k)}(\lambda)$ in that series turns out to be a polynomial in the subsystem size $\ell$ of order $k$, which eventually diverges as $\ell \to \infty$. Moreover, the resulting expansion of the probability distribution function, i.e.

$$
\widetilde{P}_\ell(m) = \sum_{k=0}^{\infty} \widetilde{P}_\ell^{(2k)}(m) h^{2k}, \quad (57)
$$

is such that each term $\tilde{P}_\ell^{(2k)}(m)$ is different from zero only for $m \in [-\ell/2, \cdots -\ell/2+k] \cup [\ell/2-k, \ldots, \ell/2]$. For example, up to the fourth order one easily obtains

$$
\begin{aligned}
\widetilde{P}_\ell^{(0)}(m) &= \frac{1}{2}(\delta_{m,-\ell/2} + \delta_{m,\ell/2}), \quad (58) \\
\widetilde{P}_\ell^{(2)}(m) &= -\frac{\ell}{32}(\delta_{m,-\ell/2} - \delta_{m,-\ell/2+1} - \delta_{m,\ell/2-1} + \delta_{m,\ell/2}), \\
\widetilde{P}_\ell^{(4)}(m) &= \frac{\ell(\ell-9)-6}{1024}(\delta_{m,-\ell/2} - \delta_{m,-\ell/2+1} - \delta_{m,\ell/2-1} + \delta_{m,\ell/2}) \\
&\quad - \frac{(\ell-1)(\ell+6)}{1024}(\delta_{m,-\ell/2+1} - \delta_{m,-\ell/2+2} - \delta_{m,\ell/2-2} + \delta_{m,\ell/2-1}). \quad (59)
\end{aligned}
$$

Notice that the previous expansion is valid as far as the full probability remains non-negative and smaller than one; therefore, the result up to the forth order is meaningful only for $\ell \lesssim 2 + 16/h^2$.

## 5.2 Scaling limit at fixed $h^2\ell$

We have seen from the previous section that the truncated series (52) is not accurate for arbitrary $\ell$. Since we may be interested in the asymptotic behaviour for $\ell \gg 1$ and $h \ll 1$, we can exploit the fact that

$$
F_\ell^{(2k)}(\lambda) = \sum_{j=0}^{k} f_j^{(2k)}(\lambda, \ell) \ell^j, \quad (60)
$$

where the coefficient are complicated combinations of trigonometric functions such that $|f_j^{(2k)}(\lambda, \ell)| < 1/j!$ for $\lambda \in [-\pi, \pi]$ and arbitrary $\ell$. Within this working hypothesis, when considering the limit $h \to 0$ and $\ell \gg 1$ keeping constant $\mathfrak{h} \equiv h\sqrt{\ell}$, only the highest order term in the polynomial expansion of $F_\ell^{(2k)}(\lambda)$ contributes, and the generating function reduces to

$$
F_\ell(\lambda) \sim \sum_{k=0}^{\infty} f_k^{(2k)}(\lambda, \ell) \mathfrak{h}^{2k} = e^{-\mathfrak{h}^2 \sin(\lambda/2)^2/8} \cos[\ell\lambda/2 - \mathfrak{h}^2 \sin(\lambda)/16], \quad (61)
$$

where the last equality has been checked by comparing the r.h.s series expansion against the exact coefficient; in particular we were able to verify that

$$
f_k^{(2k)}(\lambda, \ell) = \frac{\sin(\lambda/2)^k}{8^k k!} \cos[(\ell-k)\lambda/2 - k\pi/2], \quad (62)
$$

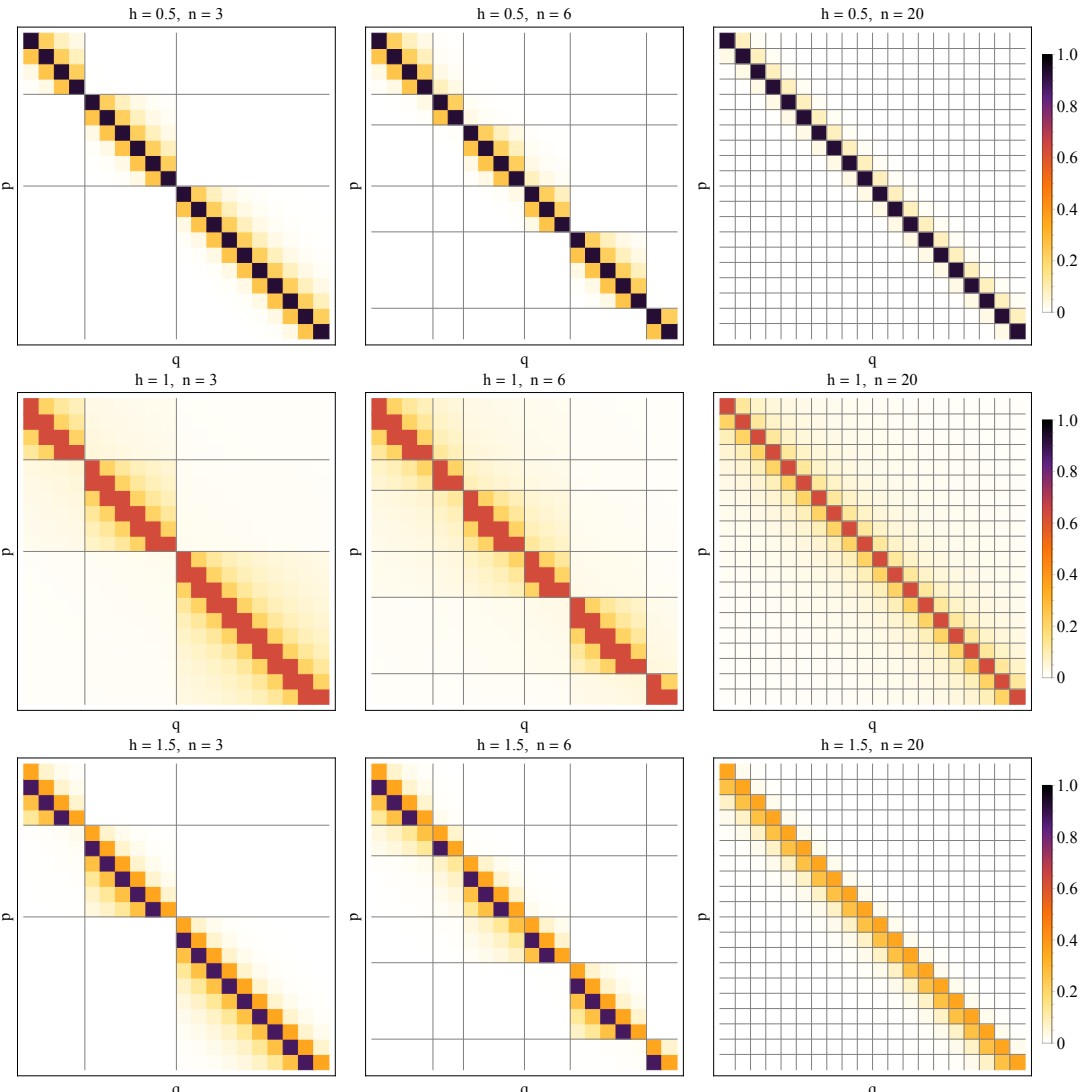

Figure 5: Matrix density plot of the absolute value of $\mathbb{G}_{j_n}$ for $\ell = 40$ and different value of $h$ and $n$. The straight grey lines define the sub-intervals $[j_{2i-1}, j_{2i}]$ in order to highlight the block structure of the matrices. (**left column**) $n = 3$ and $j_3 = \{1, 5, 10, 16, 30, 40\}$. (**center column**) $n = 6$ and $j_6 = \{1, 5, 6, 8, 10, 14, 15, 18, 27, 32, 38, 40\}$. (**right column**) $n = 20$ and $j_{20} = \{i : i \in [1, 40]\}$. Notice that the intervals have been chosen so as all matrices have the same dimension $\mathcal{L}_{j_n} = \sum_{i=1}^{n} (j_{2i} - j_{2i-1}) = 20$.

up to $k = 2$. In this scaling regime, the emptiness formation probability takes the simple form $\mathcal{E}_\ell \sim \exp(-\hbar^2/16)/2$. Further support to Eq. (61) will come later on, in the framework of the block-diagonal approximation, when expanding Eq. (70) and keeping $\hbar$ constant (*c.f.* Sec. 5.3). Finally, let us mention that this scaling limit will turn out to be very useful to understand the stationary probability distribution we found in the previous section so as to point out a strict relation with the ground state properties (*c.f.* Sec. 6).

### 5.3 Block-diagonal approximation

A more effective approximation of the full counting statistics of the order parameter in the ground-state may rely on the block structure of the matrix $\mathbb{G}_{j_n}$. As a matter of fact the entries of this matrix are given by $g_{p-q}$ where $p$ and $q$ move in the union of intervals $\bigcup_{i=1}^{n} \mathcal{I}_i$, where $\mathcal{I}_i = [j_{2i-1}, j_{2i}-1]$ with $1 \le j_1 < j_2 < \cdots < j_{2n} \le \ell$. Explicitly one can write

$$\mathbb{G}_{j_n} = \begin{bmatrix} \mathbb{G}_{\mathcal{I}_1,\mathcal{I}_1} & \mathbb{G}_{\mathcal{I}_1,\mathcal{I}_2} & \cdots & \mathbb{G}_{\mathcal{I}_1,\mathcal{I}_n} \\ \mathbb{G}_{\mathcal{I}_2,\mathcal{I}_1} & \mathbb{G}_{\mathcal{I}_2,\mathcal{I}_2} & \cdots & \mathbb{G}_{\mathcal{I}_2,\mathcal{I}_n} \\ \vdots & \vdots & \ddots & \vdots \\ \mathbb{G}_{\mathcal{I}_n,\mathcal{I}_1} & \mathbb{G}_{\mathcal{I}_n,\mathcal{I}_2} & \cdots & \mathbb{G}_{\mathcal{I}_n,\mathcal{I}_n} \end{bmatrix}, \tag{63}$$

where $\mathbb{G}_{\mathcal{I}_p,\mathcal{I}_q}$ is a $|\mathcal{I}_p| \times |\mathcal{I}_q|$ matrix with entries

$$\left(\mathbb{G}_{\mathcal{I}_p,\mathcal{I}_q}\right)_{\alpha,\beta} = g_{\alpha-\beta+(j_{2p-1}-j_{2q-1})}, \quad \alpha \in [1,|\mathcal{I}_p|], \quad \beta \in [1,|\mathcal{I}_q|]. \tag{64}$$

Since $g_n$ is in general decaying with $n$, the last relation confirms the fact that, as far as we consider off-diagonal blocks connecting two far apart intervals, their contribution to the determinant may be "sub-leading". In particular, for $h$ sufficiently far from the critical point $g_n$ decays exponentially, and this naïve argument should be more effective.

In Figure 5 we show matrix density plots of the absolute value of $\mathbb{G}_{j_n}$ for $\ell = 40$, $h = \{0.5, 1, 1.5\}$ and three different configurations of $\boldsymbol{j}_n$. As expected, the matrices take on their larger values into the block-diagonal sectors, whereas the off-diagonal blocks are almost vanishing; at least as far as $h$ is sufficiently far from the critical value. However, when $n$ is such large that sub-intervals are contiguous (like the rightmost case in Figure 5), then off-diagonal blocks may be no longer negligible even though they are anyhow smaller than the corresponding diagonal entries.

For these reasons, we decided to split matrix $\mathbb{G}_{j_n}$ in block-diagonal and off-diagonal-block terms

$$\mathbb{G}_{j_n} = \mathbb{D}_{j_n} + \mathbb{X}_{j_n} = \mathbb{D}_{j_n}[\mathbb{I} + \mathbb{D}_{j_n}^{-1}\mathbb{X}_{j_n}], \tag{65}$$

with

$$\mathbb{D}_{j_n} \equiv \begin{bmatrix} \mathbb{G}_{\mathcal{I}_1,\mathcal{I}_1} & \mathbb{0} & \cdots & \mathbb{0} \\ \mathbb{0} & \mathbb{G}_{\mathcal{I}_2,\mathcal{I}_2} & \cdots & \mathbb{0} \\ \vdots & \vdots & \ddots & \vdots \\ \mathbb{0} & \mathbb{0} & \cdots & \mathbb{G}_{\mathcal{I}_n,\mathcal{I}_n} \end{bmatrix}, \quad \mathbb{X}_{j_n} \equiv \begin{bmatrix} \mathbb{0} & \mathbb{G}_{\mathcal{I}_1,\mathcal{I}_2} & \cdots & \mathbb{G}_{\mathcal{I}_1,\mathcal{I}_n} \\ \mathbb{G}_{\mathcal{I}_2,\mathcal{I}_1} & \mathbb{0} & \cdots & \mathbb{G}_{\mathcal{I}_2,\mathcal{I}_n} \\ \vdots & \vdots & \ddots & \vdots \\ \mathbb{G}_{\mathcal{I}_n,\mathcal{I}_1} & \mathbb{G}_{\mathcal{I}_n,\mathcal{I}_2} & \cdots & \mathbb{0} \end{bmatrix},$$

in order to expand the logarithm of the determinant of each matrix $\mathbb{G}_{j_n}$ as

$$\log[\det(\mathbb{G}_{j_n})] = \sum_{p=1}^{n} \log[\det(\mathbb{G}_{\mathcal{I}_p,\mathcal{I}_p})] - \sum_{k=1}^{\infty} \frac{(-1)^k}{k} \mathrm{Tr}\left[\left(\mathbb{D}_{j_n}^{-1}\mathbb{X}_{j_n}\right)^k\right],$$

where the first non-vanishing correction to the block-diagonal result is order $k = 2$ since for symmetry reason one has $\mathrm{Tr}(\mathbb{D}_{j_n}^{-1}\mathbb{X}_{j_n}) = 0$. Of course, the previous expansion relies on the fact that the trace of the matrix power $(\mathbb{D}_{j_n}^{-1}\mathbb{X}_{j_n})^k$ decays as $k$ become larger. We have numerical evidence that this is the case.

However, the evaluation of the generating function (31) requires, for each $n \in [0, \lfloor \ell/2 \rfloor]$, a sum over all possible admissible configurations of the indices $\boldsymbol{j}_n$. Unfortunately, this means that corrections due to the off-diagonal blocks for a specific configuration of indices $\boldsymbol{j}'_{n'}$ with $n'$ sufficiently large may be of the same order of the leading block-diagonal contribution coming from a different configuration $\boldsymbol{j}''_{n''}$ with $n''$ much smaller than $n'$. From simple counting

arguments (see for example (55)), we expect that the number of configurations with $n$ close to $\ell$ are exponentially suppressed, therefore we may loosely approximate the ground-state generating function by keeping only the block-diagonal contribution

$$F_\ell(\lambda) \simeq \cos(\lambda/2)^\ell \sum_{n=0}^{\lfloor \ell/2 \rfloor} [i \tan(\lambda/2)]^{2n} \sum_{j_1 < j_2 < \cdots < j_{2n}} \mathcal{D}_{j_2-j_1} \mathcal{D}_{j_4-j_3} \cdots \mathcal{D}_{j_{2n}-j_{2n-1}}, \tag{66}$$

where we defined $\mathcal{D}_z \equiv \det(\mathbb{G}_{[1,z],[1,z]}) = \langle \sigma_1^x \sigma_{1+z}^x \rangle$ (for $z > 0$) and we exploited the translational invariance of the block-diagonal matrices. The approximate representation (66) basically requires the evaluation of a multidimensional discrete convolution, and it can be easily computed by introducing the following discrete Fourier transforms

$$\widetilde{\mathcal{D}}(\omega) = \sum_{z=1}^{\ell} \mathcal{D}_z e^{-i\omega z}, \quad \widetilde{\theta}(\omega) = \sum_{z=1}^{\ell} e^{-i\omega z} = \frac{e^{-i\ell\omega} - 1}{1 - e^{i\omega}}, \tag{67}$$

thus obtaining

$$
\begin{aligned}
F_\ell(\lambda) &\simeq \cos(\lambda/2)^\ell \left\{ 1 + \sum_{n=1}^{\lfloor \ell/2 \rfloor} [i\tan(\lambda/2)]^{2n} \int_{-\pi}^{\pi} \frac{d\omega}{2\pi} \widetilde{\theta}(-\omega)[\widetilde{\theta}(\omega)\widetilde{\mathcal{D}}(\omega)]^n \right\} \\
&= \cos(\lambda/2)^\ell \left\{ 1 + \int_{-\pi}^{\pi} \frac{d\omega}{2\pi} \widetilde{\theta}(-\omega) \frac{[-\tan(\lambda/2)^2 \widetilde{\theta}(\omega)\widetilde{\mathcal{D}}(\omega)]^{\lfloor \ell/2 \rfloor} - 1}{[\tan(\lambda/2)^2 \widetilde{\theta}(\omega)\widetilde{\mathcal{D}}(\omega)]^{-1} + 1} \right\}. \tag{68}
\end{aligned}
$$

Last equation has the great advantage of having recasted a sum of an exponentially large number of terms into a single integral in a finite support which can be evaluated with arbitrary precision[2]. Thereafter, by using Eq. (68) into Eq. (14), we obtain an approximative description for the ground-state PDF. This is the leading result of this section and it turns out the be very accurate as far as the state is characterised by a short correlation length.

In Figure 6 we plot the ground-state probability distribution function of the subsystem longitudinal magnetisation for different values of the transverse field $h$ and subsystem sizes $\ell$. We compare the exact result obtained from the direct evaluation of Eq. (31) against the block-diagonal approximation given by using Eq. (68). As far as the system is sufficiently far from the critical point, the approximation works very well. We expect that the validity domain of the approximation scales with the value of the transfer field, and becomes larger as $h$ moves away from the critical value.

Interestingly Eq. (66) may be further simplified for $\ell \gg 1$ if one assumes that the largest contributions to the sum come from the asymptotic expansion of the two-point correlation function. This can be easily worked out both in the ferromagnetic ($0 \le h < 1$) and paramagnetic ($h > 1$) phases separately.

- For $0 \le h < 1$, Szëgo theorem leads to the following asymptotic expansion of the two point correlation function [61]

$$\mathcal{D}_z \simeq (1-h^2)^{1/4} \equiv 4\mu_\pm^2, \tag{69}$$

where $\mu_+$ ($\mu_-$) is the order parameter expectation value in the symmetry-broken phase with positive (negative) magnetisation. Using this result in (66) we simply obtain

$$F_\ell(\lambda) \simeq \frac{1}{2} \left[ \cos(\lambda/2) + i\,2\mu_\pm \sin(\lambda/2) \right]^\ell + \frac{1}{2} \left[ \cos(\lambda/2) - i\,2\mu_\pm \sin(\lambda/2) \right]^\ell, \tag{70}$$

---

[2] Notice that Eqs. (66) and (68) turned out to be exact when describing the stationary distribution after the melting of the ferromagnetic order (see Sec. 4).

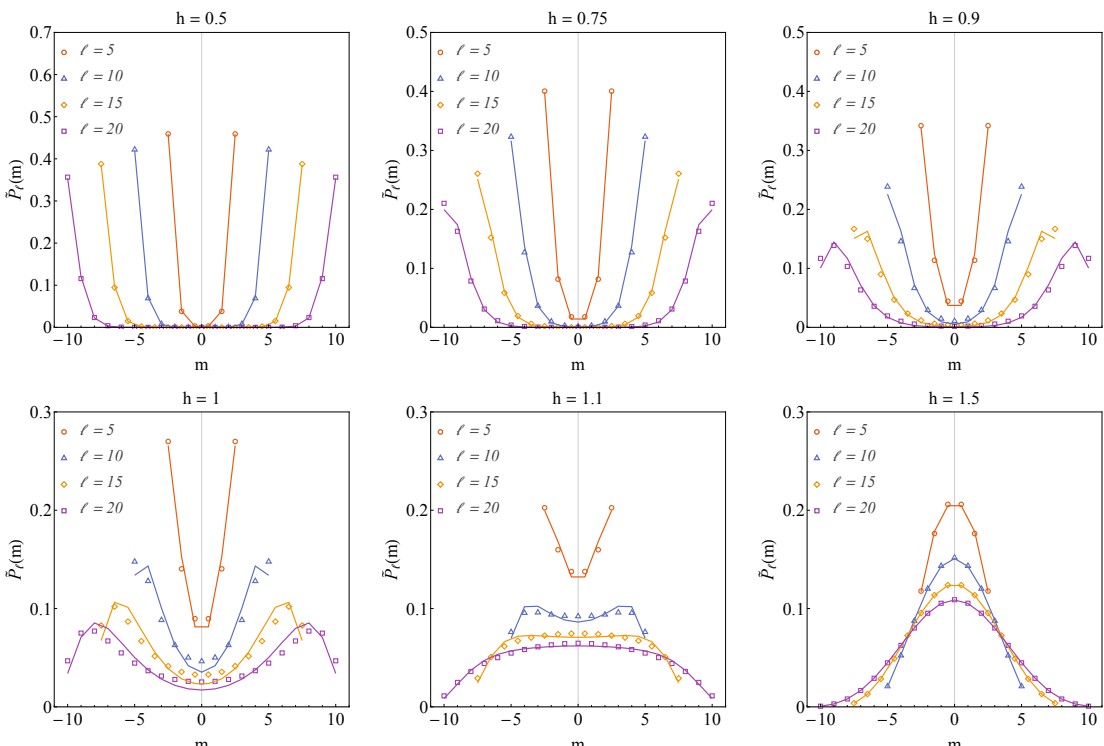

Figure 6: Exact ground-state PDF (symbols) is compared with the block diagonal approximation obtained by using Eq. (68) (full lines) for relatively small subsystem sizes and different values of the transverse field.

which, for $\ell \gg 1$, leads to the following asymptotic scaling of the PDF

$$\widetilde{P}_\ell(m) \simeq \frac{1}{\sqrt{2\pi\ell(1/4-\mu_\pm^2)}} \exp\left[-\frac{m^2+\ell^2\mu_\pm^2}{2\ell(1/4-\mu_\pm^2)}\right] \cosh\left[\frac{\mu_\pm m}{1/4-\mu_\pm^2}\right], \qquad (71)$$

which, as expected, coincides with the superposition of two asymptotic Gaussian distribution (20) with variance

$$\sigma^2 = \lim_{\ell\to\infty} \frac{1}{\ell}\langle\mu_\pm|M_\ell^2|\mu_\pm\rangle_c \simeq \frac{1}{4} - \mu_\pm^2, \qquad (72)$$

and average $\mu_\pm = \pm(1-h^2)^{1/8}/2$.

In Figure 7 we compare the exact numerical data obtained from iTEBD simulations with the bimodal approximation in Eq. (71). As expected, the agreement is better for larger subsystem sizes and smaller values of the transverse field. Notice that, Eq. (71) is not suitable for evaluating the emptiness formation probability since it has been obtained by taking the Fourier transform of the generating function in the asymptotic limit $\ell \to \infty$ with $\lambda\sqrt{\ell}$ constant. Nevertheless, we can extract the EFP directly by plugging Eq. (70) into Eq. (25), thus obtaining

$$\mathcal{E}_\ell \simeq \frac{1}{2}\left[\frac{1+(1-h^2)^{1/8}}{2}\right]^\ell + \frac{1}{2}\left[\frac{1-(1-h^2)^{1/8}}{2}\right]^\ell, \qquad (73)$$

which, in the scaling limit $h \to 0$ with constant $\mathfrak{h} = h\sqrt{\ell}$, unveils the scaling behaviour $\mathcal{E}_\ell \sim \exp(-\mathfrak{h}^2/16)/2$. Remarkably, in the same scaling hypothesis, the entire generating

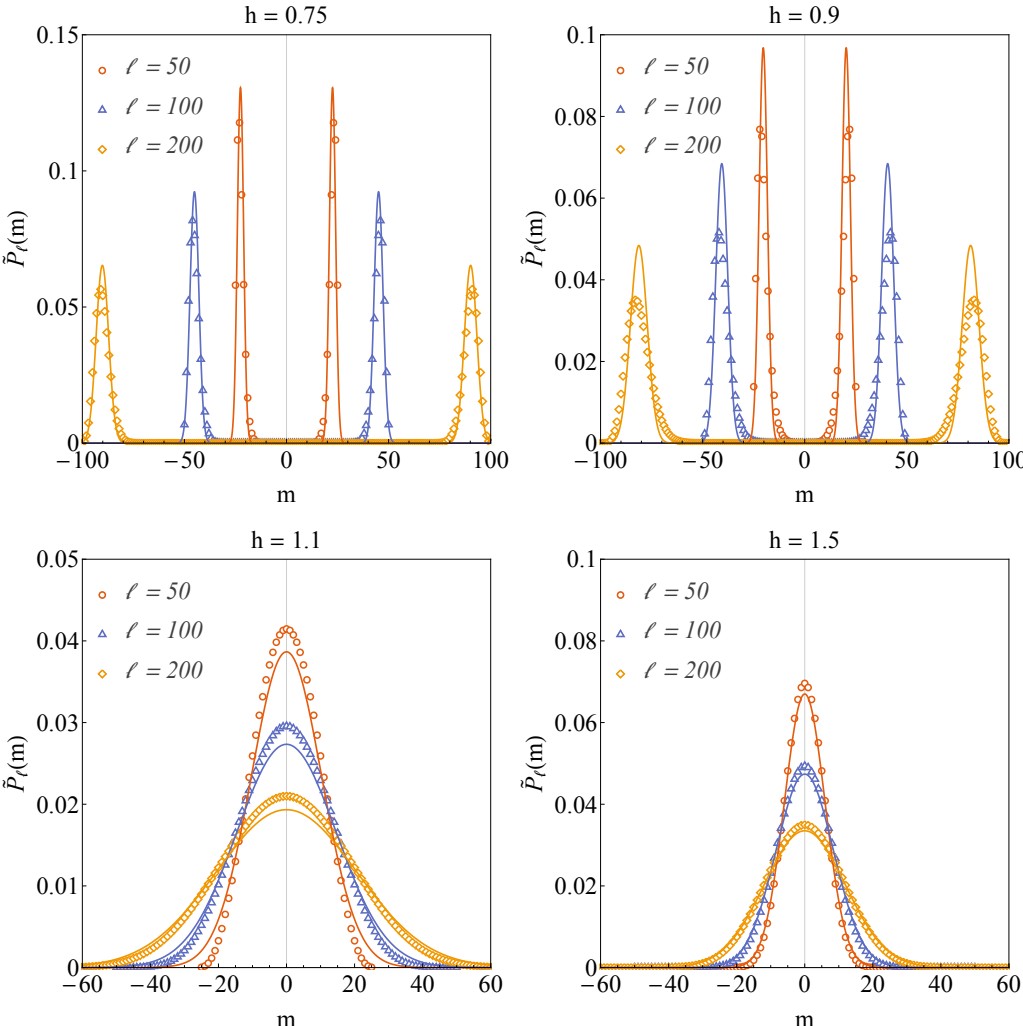

Figure 7: Asymptotic behaviour of ground-state PDF in the ferromagnetic region (top row) and in the paramagnetic region (bottom row). Symbols are the exact iTEBD results, full lines are the asymptotic approximation (71) or (75) depending on the case.

function (70) reduces to formula (61), further supporting the validity of the scaling expansion.

• In the paramagnetic phase the asymptotic behaviour of the determinant is [61]

$$\mathcal{D}_z \simeq \frac{h^{-z}}{\sqrt{\pi} z^{1/2}} \left(1 - \frac{1}{h^2}\right)^{-1/4}. \tag{74}$$

In particular, due to the presence of an exponential cut-off in the two-point function, we may directly apply the asymptotic Gaussian approximation of the PDF (see Sec. 3.1), thus having

$$\widetilde{P}_\ell(m) \simeq \frac{1}{\sqrt{2\pi\ell\sigma^2}} \exp\left[-\frac{m^2}{2\ell\sigma^2}\right], \tag{75}$$

where, for $\ell \gg 1$, the variance is given by

$$\sigma^2 \simeq \frac{1}{4} + \frac{1}{2\ell} \sum_{z=1}^{\ell-1} (\ell - z) \mathcal{D}_z \simeq \frac{1}{4} + \frac{\mathrm{Li}_{1/2}(1/h)}{2\sqrt{\pi}} \left(1 - \frac{1}{h^2}\right)^{-1/4}. \tag{76}$$

In Figure 7 we compare the exact numerical data obtained from iTEBD simulations with the Gaussian approximation. As expected, the agreement is better for larger subsystem sizes and higher values of the transverse field.

Let us finally mention that, exactly at the critical point ($h = 1$) the asymptotic behaviour of the two-point correlation function is known as well, being $\mathcal{D}_z \simeq \sqrt{\pi} G(1/2)^2 z^{-1/4}$, where $G(x)$ is the Barnes $G$-function [61]. An approximate description of the full counting statistics is hard to evaluate due to such power-law decay of the two-point function. However, the structure in Eq. (66) joined with the asymptotic behaviour of $\mathcal{D}_z$ is reminiscent of the partition function of a Coulomb gas with logarithmic interactions. In Ref. [56] the same analogy as been pointed out, and the order parameter partition function at the critical point has been related to the anisotropic Kondo problem, and its scaling form has been explicitly obtained.

# 6 Memories of the initial order

Here we try to keep a general point of view and highlight a more fundamental explanation which gives a very physical understanding behind the peculiar behaviour of the order parameter statistics in the steady-state, and explains why some remnants of the original long-range order may persist in the long-time limit after the quench. In the following discussion, although we will be referring to the specific case of the Ising quantum chain, we argue that the general arguments should be valid for any quantum system which exhibits similar characteristics, as it has been already put forward in Ref. [33].

In order to keep the discussion as general as possible, let us start from the well established fact that, after a global quantum quench, the stationary state is characterised by a finite correlation length $\xi$ (*c.f.* Appendix B). Therefore, whenever we want to measure a local observable, let us say in a subsystem of size $\ell$, we may consider the subsystem as it was the entire system, so that our measurement will eventually be affected by finite-size correction $O(\xi/\ell)$. Now it is clear that, when $\xi \lesssim \ell$, we may approximate the stationary density matrix of the full system with the stationary matrix of an analogous system with size $\ell$. From now on we are assuming this working hypothesis and all expectation values are intended on a system of finite dimension $\ell$.

For integrable systems, the stationary state is locally described in terms of the Generalised Gibbs Ensemble (GGE) [22], $\propto \exp[-\sum_j q_j Q_j]$, which is built using an infinite set of local conserved charges $[Q_j, H_\ell] = 0$; here the subscript $\ell$ is indicating that we are working on a finite system. In the specific case of the transverse field Ising quantum chain, we may use the fermionic occupation numbers $n_k \equiv \alpha_k^\dagger \alpha_k$ which are linearly related to the local charges $Q_j$ [62], so that the GGE can be rewritten as $\varrho_\ell \equiv \exp\left[-\sum_k \beta_k n_k\right]/Z$. The Lagrange multipliers $\beta_k$ (associated to $n_k$) are fixed in such a way that $\mathrm{Tr}(n_k \varrho_\ell) = \langle \Psi_0 | n_k | \Psi_0 \rangle$, and the normalisation is given by $Z = \prod_k [1 + \exp(-\beta_k)]$.

By an abuse of notation, it is clear that the stationary probability distribution function may be approximated as follow

$$P_\ell(m) = \mathrm{Tr}[\delta(M_\ell - m)\varrho_\infty] \simeq \mathrm{Tr}[\delta(M_\ell - m)\varrho_\ell] = \sum_n \gamma_{\ell,n} P_\ell(m|E_{\ell,n}), \tag{77}$$

where in the last passage we evaluated the trace over the eigenvectors $|E_{\ell,n}\rangle$ of the post-quench Hamiltonian $H_\ell$. In particular, we defined $P_\ell(m|E_{\ell,n}) \equiv \langle E_{\ell,n} | \delta(M_\ell - m) | E_{\ell,n} \rangle$ as the

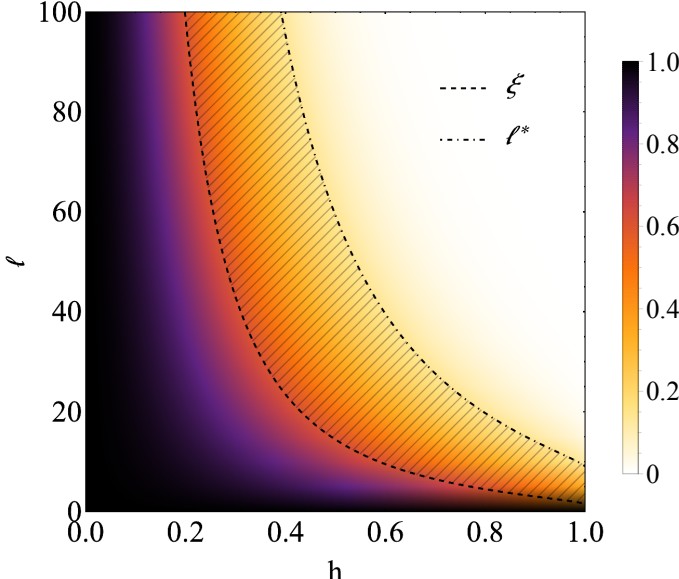

Figure 8: Density plot of the leading contribution to the ground-state overlap $\zeta_0^{2\ell}$ for different sizes and values of the magnetic field. The grey-dashed-shaded area between the correlation length $\xi$ (*c.f.* Eq. (90)) and the threshold length $\ell^* = -1/\log\zeta_0$ represents the crossover region wherein the initial long-range order and the Gaussian restoration compete so that the stationary PDF may be well approximated by Eq. (80).

conditional probability of obtaining the value $m$ when measuring $M_\ell$ provided that the state $|E_{\ell,n}\rangle$ has been fixed. Notice that, in general $P_\ell(m|E_{\ell,n}) \neq P_\ell(m|E_{\infty,n})$, for the same reason that $\varrho_\ell \neq \varrho_\infty$, indeed corrections may be important whenever $\ell \sim \xi$. Since $[\rho_\ell, H_\ell] = 0$, we also introduced the stationary overlaps $\gamma_{\ell,n}$ such that $\varrho_\ell|E_{\ell,n}\rangle = \gamma_{\ell,n}|E_{\ell,n}\rangle$. The overlaps are expected to be exponentially small in the system size and admit the asymptotic expansion $\log\gamma_{\ell,n} = \ell\log\zeta_n^2 + o(\ell)$ with $\zeta_n^2 \in [0,1]$.

Interestingly, for any finite $\ell$, whenever in Eq. (77) one particular overlap is dominating the sum, the corresponding conditional probability will definitely play a crucial role in the behaviour of the stationary PDF. It turns out that, when quenching the full ordered initial state $|\Psi_0\rangle$ within the broken-symmetry phase, the biggest contribution to the stationary probability comes from the ground-state energy-sector which is protected by a gap from the continuum excitations. As pointed out in Ref. [33], this phenomenon is not driven in any means by integrability, which simply enters in the explicit form of $\varrho_\ell$ and its eigenvalues. Also in this case, a fundamental role is played by $\zeta_0^2$ which, in the Ising quantum chain, thanks to $n_k|E_{\ell,0}\rangle = 0$, reads

$$\zeta_0^2 = \exp\left\{\int_{-\pi}^{\pi}\frac{dk}{2\pi}\log\left[1 - n_0(k)\right]\right\}, \tag{78}$$

with

$$n_0(k) \equiv \langle\Psi_0|n_k|\Psi_0\rangle = \frac{1}{2} + \frac{h\cos(k) - 1}{\epsilon_k}, \tag{79}$$

being the mode occupation function in the initial state.

Now, it is understood that, when $\ell \to \infty$, the stationary probability distribution function should acquire a Gaussian shape which is only fixed by the large deviation scaling in

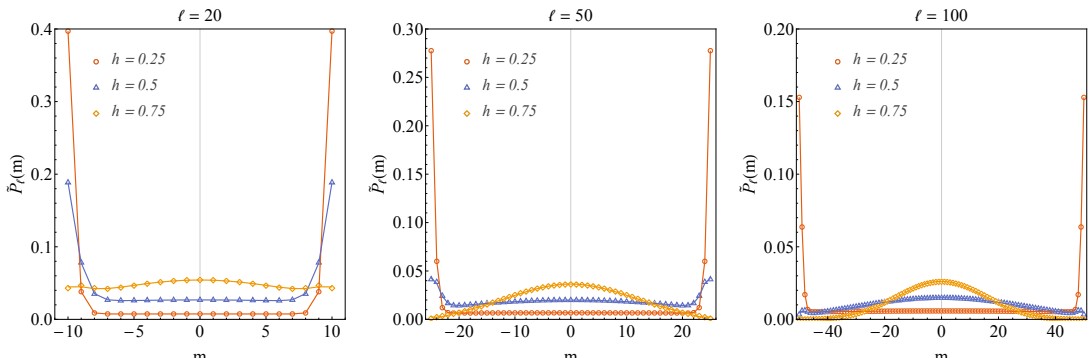

Figure 9: Exact stationary PDF obtained by Fourier transforming Eq. (48) (symbols) is compared with the best-fit phenomenological description given by Eq. (80) (full lines).

Eq. (20). Nevertheless, as far as $\zeta_0^2$ is very close to 1, which is the case for quenches which remain deep into the broken-symmetry phase, we need very large system sizes, of the order of $\ell^* = -1/\log \zeta_0$, in order to see the effects of the central limit theorem and thus Gaussian restoration (see Figure 8). Therefore, whenever $\xi \lesssim \ell \lesssim \ell^*$ (with both $\xi$ and $\ell^*$ depending on the quench parameter), Gaussian fluctuations compete with the post-quench ground-state order, and we may expect the following phenomenological behaviour for the stationary PDF[3]

$$\widetilde{P}_\ell(m) \simeq \gamma \widetilde{P}_\ell(m|E_{\infty,0}) + (1-\gamma)\widetilde{P}_\ell(m)_{Gauss}, \tag{80}$$

where $\widetilde{P}_\ell(m|E_{\infty,0})$ is the discrete PDF evaluated in the ground-state of the post-quench Hamiltonian and

$$\widetilde{P}_\ell(m)_{Gauss} \equiv \frac{\exp[-m^2/(2\delta)]}{\sum_{m=-\ell/2}^{\ell/2} \exp[-m^2/(2\delta)]} \tag{81}$$

accounts for the Gaussian fluctuations. Notice that Gaussian distribution has been normalised in such a way that $\sum_{m=-\ell/2}^{\ell/2} \widetilde{P}_\ell(m) = 1$. Here the parameter $\gamma$, which is expected to scale as $\zeta_0^{2\ell}$ for $\ell \to \infty$, nontrivially depends on the expectation value of the reduced stationary matrix in the the post-quench ground state and can be adjusted, together with the variance $\delta$ of the Gaussian fluctuations, in order to optimise the phenomenological description.

For the Ising quantum chain, in Figure 8 we represent a region of the $h$-$\ell$ plane where $h$ moves in the ferromagnetic phase and the equilibrium PDF is indeed affected by the ground-state ferromagnetic order. In Figure 9 we compare the exact stationary PDF with the phenomenological approximation in Eq. (80) where, for each $h$ and $\ell$, the parameter $\gamma$ and the variance $\delta$ of the Gaussian fluctuations have been fixed by optimising the fit[4]. The agreement between the exact stationary probability distribution function and the qualitative description is extremely good and it is expected to become even better for larger subsystem sizes and for quenches deep in the broken-symmetry phase.

As a matter of fact, Eq. (80) can be rewritten at the level of the generating function. In particular, by inspecting the exact result of the previous section, when quenching deep in the ferromagnetic phase, the phenomenological description turns out to be exact. Indeed, in the

---

[3] Here we are assuming that only one ground state contributes to the stationary probabilities as it is the case in the Ising quantum chain whenever we are confined into one symmetry sector (e.g. with $P = +1$.)

[4] We have performed a non-linear fit in $\{\gamma, \delta\}$ using Eq.(80) where the ground-state PDF fitting function $P_\ell(m|E_{\infty,0})$ is parameter-free and it has been obtained by interpolating the exact iTEBD datasets.

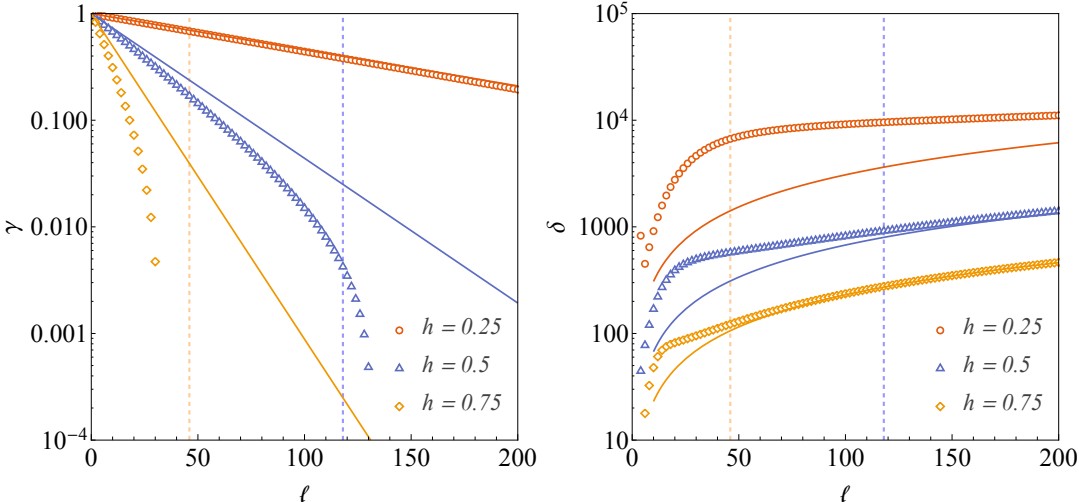

Figure 10: Logarithmic plot of the best fit parameters $\{\gamma, \delta\}$ (symbols) when using Eq. (80) to approximate the behaviour of the exact stationary PDF obtained in Sec. 4.1.2. Vertical dashed lines represent $\ell^* \simeq \{118, 46\}$ respectively for $h = \{0.5, 0.75\}$. Notice that for $h = 0.25$ the threshold size $\ell^* \simeq 503$ is outside the scale of the plots. The parameter $\gamma$ is compared with the scaling prediction $\exp(-\ell h^2/8)$ (full lines) which is expected to be a good approximation only for $h \to 0$. Similarly, the variance $\delta$ is compared with the large deviation prediction $\ell(-5/4 + 2/h^2)$ (full lines) which is expected to be valid only for $\ell \gtrsim \ell^*$.

scaling limit $h \ll 1$ and $\ell \gg 1$ with constant $\mathfrak{h} = h\sqrt{\ell}$, the stationary generating function in Eq. (48) reduces to

$$
\begin{aligned}
F_\ell(\lambda) \;\simeq\; & e^{-\mathfrak{h}^2/8}\, e^{-\mathfrak{h}^2 \sin(\lambda/2)^2/8} \cos[\ell\lambda/2 - \mathfrak{h}^2 \sin(\lambda)/16] \\
& + \left(1 - e^{-\mathfrak{h}^2/8}\right) \exp(-\ell\sigma^2\lambda^2/2),
\end{aligned}
\tag{82}
$$

where we may therefore identify, in the same scaling regime, $\gamma = \zeta_0^{2\ell} \simeq \exp(-\mathfrak{h}^2/8)$ and the large deviation variance $\sigma^2 = (-5/4 + 2/h^2)$. In the previous exact scaling formula the first line contribution matches the ground-state contribution we have reported in Eq. (61). Let us stress here that Eq. (82), and the associated probability distribution function, are parameter-free and they are expected to reproduce very well the thermodynamic behaviour deep in the broken-symmetry phase. Moreover, this result goes beyond the small-$h$ expansion (at fixed $\ell$) of the stationary generating function in Eq. (48) and relies on the explicit scaling expression of the ground-state Ising overlap and generating function[5].

In Figure 10 we show the best-fit parameters $\gamma$ and $\delta$ as a function of the subsystem size $\ell$ for different values of the post-quench magnetic field $h$. As expected, whenever $\ell \gtrsim \ell^*$, $\gamma \simeq 0$ and the Gaussian behaviour is fully restored, with $\delta$ which is well approximated by the large deviation scaling $\ell\sigma^2 = \ell(-5/4 + 2/h^2)$. Similarly, for $h$ sufficiently small (see $h = 0.25$ in Figure 10), the parameter $\gamma$ is in good agreement with the scaling result $\exp(-\mathfrak{h}^2/8)$.

Finally, in Figure 11 we compare the stationary generating function given by Eq. (48) with the scaling behaviour in Eq. (82). As expected the agreement is better for large subsystem sizes and the approaching to the scaling formula is faster for smaller rescaled parameter $\mathfrak{h}$, i.e. deeper into the ferromagnetic phase.

---

[5]The small-$h$ expansion here would be the analogous of the $1/\Delta$ expansion of the $XXZ$ model in Ref. [33].

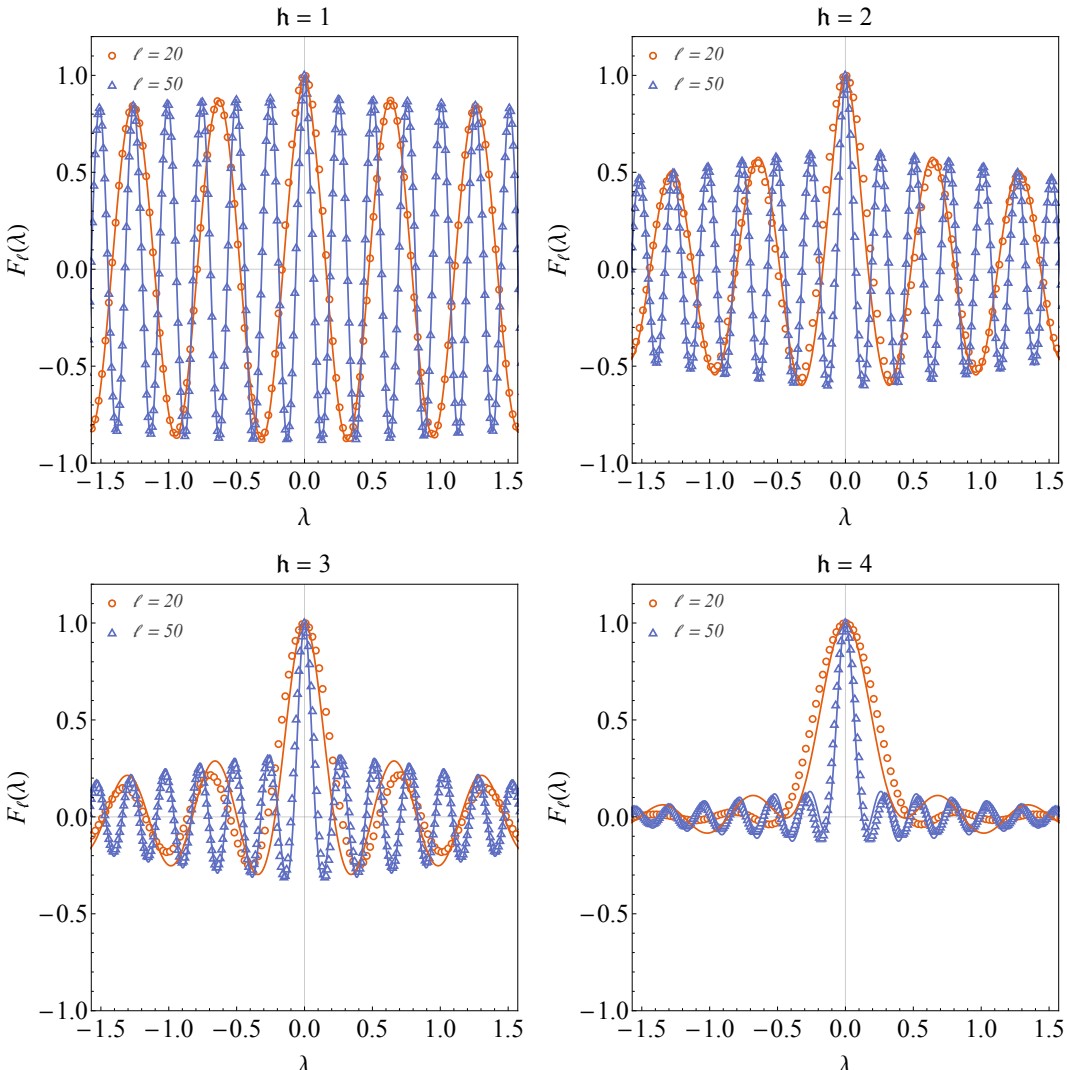

Figure 11: The exact stationary generating function in Eq. (48) (symbols) is compared with the scaling behaviour (82) (full lines) for different values of the rescaled parameter $\mathfrak{h} = \{1, 2, 3, 4\}$ which corresponds to scaling value of the parameter $\gamma = \exp(-\mathfrak{h}^2/8) \simeq \{0.8825, 0.6065, 0.3246, 0.1353\}$. For clarity reason here we plot only the region $\lambda \in [-\pi/2, \pi/2]$. Notice that in these plots we did not adjusted any parameter by fitting.

## 7 Conclusion

We have studied the full counting statistics of the local order parameter in the transverse field Ising quantum chain. At the equilibrium at zero temperature, we proposed a fairly accurate description for the corresponding generating function which is based on the diagonal approximation of the determinant representation. However, we mainly focus on the non-equilibrium dynamics of the probability distribution function when quenching the system from a fully polarised initial state (namely from $h = 0$). We were interested in characterising the melting of the local ferromagnetic order. We therefore determined the PDF in the stationary state reached at late times after the quench. Thanks to a remarkable connection with the partition function of a 3-states classical model, we were able to obtain a closed analytical description of the full

counting statistics at the late-time. In this sense, with Eq. (48), our work provides the first analytical result for the order parameter statistics at late time after a quantum quenches in the transverse field Ising quantum chain.

As expected, for any values of the post-quench field $h$ and sufficiently large subsystem sizes, the stationary full counting statistics acquires a Gaussian shape. However, when quenching within the ferromagnetic region, the PDF may exhibit a very broad bimodal shape, which is as much pronounced as deeply we quench in the broken-symmetry phase. We may say that, for any finite subsystem, the stationary state somehow keeps memories of the the long-range order encoded in the initial state.

Finally, we also provided a very general explanation of the peculiar behaviour of the order parameter statistics grounded on a more physical understanding. We enlightened a strong connection between the stationary PDF and the PDF evaluated in the ground-state of the post-quench Hamiltonian. When quenching within the broken-symmetry phase, the crossover from a bimodal distribution to a simple normal distribution, which is visible for finite subsystems, is somehow related to the thermodynamic behaviour of the overlap between the initial state and the final ground-state. With this respect, in the scaling regime $h \ll 1$, $\ell \gg 1$ with constant $\mathfrak{h} = h\sqrt{\ell}$, we obtained the exact scaling formula (82).

# 8 Acknowledgements

The author is very grateful to Andrea De Luca, Fabian Essler, Maurizio Fagotti and Simone Montangero for fruitful discussions. Fabian Essler is especially acknowledged for evaluable suggestions and for the careful reading of the manuscript. Part of this work has been carried out during the workshop "Quantum Paths" at the Erwin Schrödinger International Institute in Vienna and during the workshop "Entanglement in quantum systems" at the Galileo Galilei Institute in Florence.

**Funding information –** This work was supported by the European Union's Horizon 2020 research and innovation program under the Marie Sklodowska-Curie Grant Agreement No. 701221 NET4IQ, as well as by the BMBF and EU-Quantera via QTFLAG, and the Quantum Flagship via PASQuanS.

# A From generating function to probability distribution function

Passing from Eq. (23) to Eq. (24) requires the evaluation of the following integral

$$p_{m,n}(\ell) \equiv \int_{-\pi}^{\pi} \frac{d\lambda}{2\pi} e^{-im\lambda} \cos(\lambda/2)^{\ell-2n}[i\sin(\lambda/2)]^{2n}, \tag{83}$$

where $n \in [0, \ldots, \lfloor \ell/2 \rfloor]$ and $m \in [-\ell/2, \ldots, \ell/2]$. Due to the parity symmetry of the PDF, we have $p_{-m,n}(\ell) = p_{m,n}(\ell)$ thus we focus on $m \geq 0$. Rewriting the the exponential by means of the Binomial Theorem (since $2m$ is integers)

$$e^{-im\lambda} = [\cos(\lambda/2) - i\sin(\lambda/2)]^{2m} = \sum_{k=0}^{2m} \binom{2m}{k} \cos(\lambda/2)^k [-i\sin(\lambda/2)]^{2m-k}, \tag{84}$$

and using (for $p$ and $q$ non-negative integers)

$$\int_{-\pi}^{\pi} \frac{d\lambda}{2\pi} \cos(\lambda/2)^q \sin(\lambda/2)^p = \frac{1+(-1)^p}{2\pi} \frac{\Gamma[(1+p)/2]\Gamma[(1+q)/2]}{\Gamma[(p+q)/2+1]}, \tag{85}$$

one has

$$p_{m,n}(\ell) = \frac{(-1)^n}{\pi} \sum_{k=0}^{2|m|} \frac{2\cos[(k-2|m|)\pi/2]}{\ell+k-2n+1} \binom{2|m|}{k} \binom{\ell/2+|m|}{|m|+n-k/2-1/2}^{-1}, \qquad (86)$$

where the absolute value has been introduced in order to extend the result to $m < 0$.

## B  Stationary two-point function after quenching from $h = 0$ to $|h| \leq 1$

The stationary generating function and the related probability distribution function of the order parameter when quenching the state (32) within the ferromagnetic phase can be basically evaluated by using the analytical expression for the stationary two-point function $\mathcal{D}_z = \langle \sigma_1^x \sigma_{1+z}^x \rangle$. Indeed, we have

$$\mathcal{D}_z = \det \begin{pmatrix} g_0 & g_{-1} & g_{-2} & g_{-3} & \cdots & \cdots & g_{-z+1} \\ g_1 & g_0 & g_{-1} & g_{-2} & \cdots & \cdots & g_{-z+2} \\ 0 & g_1 & g_0 & g_{-1} & \cdots & \cdots & g_{-z+3} \\ 0 & 0 & g_1 & g_0 & \cdots & \cdots & g_{-z+4} \\ \vdots & \vdots & \vdots & \vdots & \ddots & \ddots & \vdots \\ \vdots & \vdots & \vdots & \vdots & \ddots & \ddots & g_{-1} \\ 0 & 0 & 0 & 0 & \cdots & g_1 & g_0 \end{pmatrix}, \qquad (87)$$

with $g_1 = -h/2$, $g_0 = (1-h^2/2)$, and $g_{-n} = (h^n - h^{n+2})/2$ for $n > 0$. Therefore, from the results on the determinant of Toeplitz-Hessenberg matrices [63], by introducing the analytic function

$$f(x) = \sum_{n=0}^{\infty} \frac{g_{1-n}}{g_1} x^n = \frac{h-2x+hx^2}{h-h^2x}, \qquad (88)$$

the determinant admits the following closed expression

$$\mathcal{D}_z = \frac{(-g_1)^z}{z!} \partial_x^z [f(x)^{-1}]\big|_0 = \left(\frac{1+\sqrt{1-h^2}}{2}\right)^{z+1} + \left(\frac{1-\sqrt{1-h^2}}{2}\right)^{z+1}, \qquad (89)$$

which, as expected, reduces to the simple exponential behaviour $(1/2)^z$ when $h = 1$, and trivially to 1 when $h = 0$. For large distances $z \gg 1$, only the largest term in Eq. (89) contributes, and the stationary two-point correlation function decays exponentially with typical correlation length

$$\xi \equiv -\log[(1+\sqrt{1-h^2})/2]^{-1}, \qquad (90)$$

in agreement with the asymptotic findings in Ref. [28].

## C  Partition function of the Ising chain

Here we show that Eq. (39) is related to the partition function of the 1D Ising model as reported in Eq. (40). Let us start by introducing the classical Ising Hamiltonian for a chain with $\ell + 1$ classical spins and fixed boundary conditions ($s_0 = s_\ell = -1$)

$$\mathcal{H}_I = -J \sum_{j=0}^{\ell-1} s_j s_{j+1} - h \sum_{j=0}^{\ell} s_j, \qquad (91)$$

where for $j \in \{1, \dots \ell-1\}$ $s_j = \pm 1$, $J$ represents the ferromagnetic interaction and $h$ the local magnetic field. The canonical partition function $\mathrm{Tr}[\exp(-\beta \mathcal{H}_I)]$ is therefore given by (with $\Lambda = \beta J$ and $A = \beta h$)

$$\mathcal{Z}_I(\Lambda, A, \ell+1) = \sum_{\{s_j\}} \exp\left[\Lambda \sum_{j=0}^{\ell-1} s_j s_{j+1} + A \sum_{j=0}^{\ell} s_j\right] = e^{\Lambda \ell} e^{-A(\ell+1)} \sum_{d=0}^{\ell} e^{-2\Lambda d} \sum_{n_\uparrow=0}^{\ell-1} \mathcal{N}_{d,n_\uparrow} e^{2An_\uparrow}, \quad (92)$$

where $d$ counts the number of domain walls, $n_\uparrow$ the number of spins up, and $\mathcal{N}_{d,n_\uparrow}$ is the number of configurations for a given $\{d, n_\uparrow\}$. Notice that, due to the fixed boundary conditions, the number of domain walls has to be even ($\mathcal{N}_{d,n_\uparrow} = 0$ for $d$ odd), therefore the previous sum can be rewritten as

$$\mathcal{Z}_I(\Lambda, A, \ell+1) = e^{\Lambda \ell} e^{-A(\ell+1)} \sum_{d=0}^{\lfloor \ell/2 \rfloor} (e^{-2\Lambda})^{2d} \sum_{j_1 < j_2 < \cdots < j_{2d}}^{\ell} (e^{2A})^{\mathcal{L}_{j_d}}, \quad (93)$$

where $\mathcal{L}_{j_d}$ corresponds to the number of spins up $n_\uparrow$ associated to a given configuration of domain walls $\{j_1, j_2, \dots, j_{2d}\}$. Last equation is nothing more than what have been stated in Eq. (40). Interestingly, the evaluation of the 1D Ising partition function is analytically doable by means of the Transfer Matrix approach. Indeed, one can easily show that

$$\mathcal{Z}_I(\Lambda, A, \ell+1) = e^{-A} \langle -|\mathbb{T}_I^\ell|-\rangle, \quad (94)$$

where in the fictitious spin basis $|+\rangle = \begin{pmatrix} 1 \\ 0 \end{pmatrix}$, $|-\rangle = \begin{pmatrix} 0 \\ 1 \end{pmatrix}$, we have

$$\mathbb{T}_I = \begin{pmatrix} e^{\Lambda+A} & e^{-\Lambda} \\ e^{-\Lambda} & e^{\Lambda-A} \end{pmatrix}. \quad (95)$$

Matrix $\mathbb{T}_I$ can be easily diagonalised, namely $\mathbb{T}_I|z_j\rangle = z_j|z_j\rangle$, where the two eigenvalues satisfy $z_1 + z_2 = \mathrm{Tr}(\mathbb{T}_I) = 2e^\Lambda \cosh(A)$, $z_1 z_2 = \det(\mathbb{T}_I) = 2\sinh(2\Lambda)$, thus obtaining

$$z_{1,2} = e^\Lambda \cosh(A) \pm \sqrt{e^{2\Lambda} \cosh(A)^2 - 2\sinh(2\Lambda)}. \quad (96)$$

The expectation value in (94) can be written as

$$\langle -|\mathbb{T}_I^\ell|-\rangle = \langle -|z_1\rangle\langle z_1|-\rangle z_1^\ell + \langle -|z_2\rangle\langle z_2|-\rangle z_2^\ell, \quad (97)$$

with the following boundary overlaps

$$\langle -|z_j\rangle\langle z_j|-\rangle = \frac{e^{-4\Lambda}}{e^{-4\Lambda} + (e^{-\Lambda}z_j - e^{-B})^2}. \quad (98)$$

## D  Partition function of the 3-states Potts chain

Let us consider the following 3-states Potts Hamiltonian for a chain with $\ell+1$ sites

$$\mathcal{H}_P = -\sum_{j=0}^{\ell-1} J_{s_j, s_{j+1}}(1 - \delta_{s_j, s_{j+1}}) - \sum_{j=0}^{\ell} h_{s_j}, \quad (99)$$

where $s_j \in \{a, \emptyset, b\}$ for $j \in \{1, \dots \ell-1\}$, and we set fixed boundary conditions, $s_0 = s_\ell = \emptyset$. The couplings $J_{s,s'}$ accounts for energy cost when transition $s \leftrightarrows s'$ occurs between neighbouring sites; $h_s$ is a local field which does depend on the on-site state $s$.

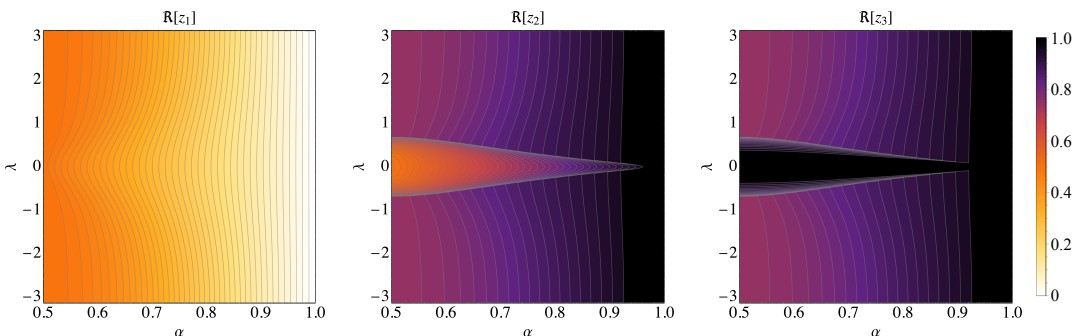

Figure 12: Contour plot of the real part of the eigenvalues $z_j$ of the transfer matrix $\mathbb{T}_P$, for $\lambda \in [-\pi, \pi]$ and post-quench parameter $\alpha \in [1/2, 1]$. Notice that, $\mathfrak{R}[z_1] \leq \mathfrak{R}[z_2] \leq \mathfrak{R}[z_3]$, with the first eigenvalue $z_1$ always real and smaller than $1/2$. The others two eigenvalues may be either reals and different, or complex conjugate, depending on the value of $\lambda$ and $\alpha$.

In order to connect the partition function of such model with the generating function in Eq. (47), we need to suppress any transition $a \leftrightarrows b$, by setting $J_{a,b} = J_{b,a} = -\infty$. Moreover, we do not have any local energy cost for the state ø, thus $h_\emptyset = 0$. Finally, when passing from ø to $a$ or $b$ and viceversa, we should pay an extra cost equal to the half of the local magnetic coupling, namely $J_{\emptyset,a} = J_{a,\emptyset} = J + h_a/2$ and $J_{\emptyset,b} = J_{b,\emptyset} = J + h_b/2$. With this choice of the couplings, the partition function $\mathrm{Tr}[\exp(-\beta \mathcal{H}_P)]$ coincides with the stationary generating function reported in Sec. 4.1.2. In particular, by exploiting the Transfer Matrix approach, we easily obtain (where $\Lambda = \beta J$, $A = \beta h_a$ and $B = \beta h_b$)

$$\mathcal{Z}_P(\Lambda, A, B, \ell + 1) = \langle \emptyset | \mathbb{T}_P^\ell | \emptyset \rangle, \tag{100}$$

where in the basis

$$|a\rangle = \begin{pmatrix} 1 \\ 0 \\ 0 \end{pmatrix}, \quad |\emptyset\rangle = \begin{pmatrix} 0 \\ 1 \\ 0 \end{pmatrix}, \quad |b\rangle = \begin{pmatrix} 0 \\ 0 \\ 1 \end{pmatrix}, \tag{101}$$

we have

$$\mathbb{T}_P = \begin{pmatrix} e^A & e^{\Lambda+A} & 0 \\ e^{\Lambda+A} & 1 & e^{\Lambda+B} \\ 0 & e^{\Lambda+B} & e^B \end{pmatrix}. \tag{102}$$

Similarly to what has been done for the Ising case, we can diagonalise the Transfer Matrix, thus obtaining

$$\langle \emptyset | \mathbb{T}_P^\ell | \emptyset \rangle = \sum_{j=1}^{3} \langle \emptyset | z_j \rangle \langle z_j | \emptyset \rangle z_j^\ell, \tag{103}$$

with $\mathbb{T}_P | z_j \rangle = z_j | z_j \rangle$ and

$$\langle \emptyset | z_j \rangle \langle z_j | \emptyset \rangle = 1 + \frac{e^{2(\Lambda+A)}}{(z_j - e^A)^2} + \frac{e^{2(\Lambda+B)}}{(z_j - e^B)^2}. \tag{104}$$

In Figure 12 we show the contour plot of the real part of the eigenvalues of the Transfer Matrix where we fixed $\Lambda = \log[i \tan(\lambda/2)]$, $A = \log(\alpha)$ and $B = \log(1 - \alpha)$ and we move $\lambda \in [-\pi, \pi]$ and $\alpha = (1 + \sqrt{1 - h^2})/2 \in [1/2, 1]$ (as in Sec. 4.1.2). As expected, depending

on the parameter region, the eigenvalues are all reals (and different), or one of them is real ($z_1$) and the other two are complex conjugate ($z_3^* = z_2$).

Notice that, in the thermodynamic limit $\ell \to \infty$, only the eigenvalues $z_3$ (with the largest modulus) will strictly contribute to the evaluation of the partition function which, as expected, will acquire a simple Gaussian shape induced by the following behaviour of the leading eigenvalue in the vicinity of $\lambda = 0$

$$\log z_3 \simeq -\left(-\frac{3}{2} + \frac{2}{h^2}\right)\frac{\lambda^2}{2}. \tag{105}$$

However, for any large but finite $\ell$, all the region $\lambda \in [-\pi, \pi]$ will turn to be important and also $z_2$ will play a crucial role, depending on the value of the quench parameter $\alpha$. This will eventually leads to a crossover in the corresponding full counting statistics: by tuning the parameter $\alpha$ the PDF will exhibit a smooth transition between a bimodal distribution and a simple normal distribution (see Figure 3).

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
