# Peer review of "Relaxation of the order-parameter statistics in the Ising quantum chain"

_SciPost Physics, doi:SciPost Phys. 7, 072 (2019)_

## Round 1 · Referee Report · Anonymous (Referee 1) · 2019-7-3

Strengths

1-tackles a not so well studied, but physically interesting problem that is the relaxation of the order-parameter statistics after quantum quenches in models with a broken-symmetry phase 2- exact results using free fermions, complemented by extensive numerics 3-nice phenomenological discussion in sec. 6

Weaknesses

1- Many typos, syntax or spelling mistakes 2- Some sloppy statements ("remarkable connection" with the 3-states Potts model sounds inexact and misleading to me... se below)

Report

The author studies the relaxation of the order parameter probability distribution function (PDF) after a quantum quench in a paradigmatic model with a spontaneously symmetry-broken phase, the transverse field Ising model.
While there is a huge amount of literature on quantum quenches, most results to this date are devoted to the study of local correlation functions or entanglement measures, and the broader question of full counting statistics has been tackled only recently. Using the free fermionic representation of the transverse field Ising model, the author re-expresses the PDF both at finite time or at equilibrium in terms of 2-point functions, which are known from previous works. This yields exact expressions for the PDFs in the late time stationary states, which allow to observe interesting physical features among which a memory of the initial order when quenching within the symmetry broken phase. The PDF within a subsystem of size $\ell$ indeed retains a bimodal structure whenever $\ell$ is not too large, while it becomes gaussian in the $\ell \to \infty$ limit.
These cute observations, supported by an interesting phenomenological comparison with equilibrium ground state order, make up for an good quality paper which deserves publication in SciPost.
However, there are several aspects (including some rather important) that I feel would need to be corrected before acceptance.

Requested changes

1 - There is a huge number of spelling or syntax mistakes, for instance :
- in the abstract : "we proof that the stationary order-parameter..." - in the intro : "doublY degenerate ground state" - later in the intro : "let us suppose to prepare" does not feel quite right - "in particular analytical findingS" - section 2, 1st sentence : "transfer field" -> "transverse field" - "Bogoliuobov" is not the most common spelling of "Bogoliubov" - before eq. (6), no spacing between "a part" - after equation (9), the use of "Indeed" seems quite improper. There is no clear logical connection between what's before and what follows - etc....
In general there are many such mistakes throughout the paper, as well as improper uses of logical connectors (some abuse of "indeed" or "However"). This should clearly be improved.

2- The introduction needs to be reorganized a little. For instance, the first paragraph deals with models with broken symmetry phases in general, so it is not fully appropriate to mention doubly degenerate ground states : either the author should mention the Ising chain (which is the focus of the paper) at this stage, or talk more generally about multiply degenerate ground states. Furthermore, in paragraph 5, the sentence "In general, after a global quantum quench, we expect long-range ferromagnetic order disappears...." sounds like a repetition of paragraph 3, and should therefore be fixed..

3- On a more physical level, it sounds quite incorrect and misleading to mention "a striking connection with the partition function of the 3-states Potts model", as is done in several places in the paper. What the authors refers to here, is the possibility to rephrase the moments generating function as the partition function of a 1D model, which can be computed through a transfer matrix approach. While this is very practical, I don't think that there is a $S_3$ symmetry here, and the only common feature with a Potts model is that one deals with a 3-states system.
If the author agrees with this, I propose to remove any reference to the Potts model, which may confuse the reader into thinking that there exists a connection between Ising and 3 states Potts models.

4- In section 6, the author spends a paragraph discussing the GGE in the transverse field Ising chain, however it is not completely clear to which extent this is used in the following. In particular, it seems that the features described here, namely a crossover as $\ell$ is increased between ordered and Gaussian behaviours, have been related in ref. [33] to features of thermal equilibrium states in a similarly integrable model. It would be useful if the author could discuss a bit the comparison of the present results and those of ref [33], and what additional features the GGE might bring.

---

## Round 1 · Referee Report · Anonymous (Referee 2) · 2019-7-21

Strengths

1- Interesting results on a not so well studied problem. 2- Careful numerical checks of all claims made.

Weaknesses

1- Clarity could be improved a bit. 2- No new analytical results for the critical point.

Report

This paper deals with the probability distribution of the order parameter in the one dimensional quantum Ising chain. Particular emphasis is put on the long time behavior after a quench from a polarized state, where analytical results are obtained using the mapping onto free fermion.

Overall the paper contains valuable results on a not so well studied problem, where the free fermion machinery is not so easy to apply. All analytical insights are supplemented by convincing numerical checks. The derivations of some of the results, as well as the writing, could be improved, however.

I list below a few issues I have with the paper, some of which are minor. The paper should be publishable in Scipost, provided those are addressed.

a) Abstract: The reference, here and elsewhere, to the three state Potts model should be removed. 'We proof' should read 'we prove'. Also, there are no rigorous proofs in the paper, so the author might consider removing such claims, and use the more neutral 'we show' instead.

b) Various typos. Page 1 'let us suppose to prepare' --> 'let us prepare'. Page 7, top: 'does not applies' --> 'does not apply'. Page 8, top 'move in the interval' --> 'are in the interval'. 'Previous equation'--> 'The previous equation'. Page 18: 'able to verified' --> 'able to verify'. There is an exclamation mark in equation (62).

c) The notation $\mathbb{I}_j$ in section 3.2 is confusing, and even leads the author to make statements that are, strictly speaking, incorrect. For a chain of length $L$ the matrix $\sigma_j^x$ is a $2^L\times 2^L$ matrix, which acts as a $2\times 2$ Pauli matrix at site $j$, and identity elsewhere. Its square or exponential is a $2^L\times 2^L$ matrix and cannot equal a $2\times 2$ matrix. It would be preferable to use $\mathbb{I}$ in formula (21) and (22), since the $2^L \times 2^L$ identity acts as identity everywhere.

d) The calculations in 4.1 are quite cryptic, and some of the expressions showed are not very readable. For example in equation (35) \sin \pi n is zero. While I understand this gets cancelled by the poles on the rhs, there surely must be a better way to present it. It might also be useful to rewrite (34) in a way that makes it clear why most Fourier coefficients are zero. Same comment around equation (51).

e) Near equation (48), it should be made clear that this is the partition function of the 1d classical Potts model, which makes the connection not particularly remarkable.

f) A reference is needed near equation (69). Also, is it the Fischer-Hartwig conjecture, or Szego's theorem? A reference is also needed around equation (74) and on page 23.

g) In several places in the paper, it is difficult to tell which results are well known, and which are new. Any improvement in that direction would be welcome.

g) It is not clear to me whether the gaussian behavior discussed in several places in the paper is an assumption that is checked, or a result that is derived.

h) There is no real discussion of the quench to the critical point $h=1$.

---

## Round 2 · Referee Report · Anonymous (Referee 1) · 2019-10-18

Report

In view of the changes made by the author I recommend it for publication in SciPost.

---

## Round 2 · Referee Report · Anonymous (Referee 1) · 2019-10-18

Report

In view of the changes made by the author I recommend it for publication in SciPost.

---

## Round 2 · Referee Report · Anonymous (Referee 2) · 2019-10-31

Report

The author has answered my questions in a satisfactory manner. I recommend publication in Scipost.

---

## Round 2 · Author Response

Dear Editor, dear referees,
I’m glad to know the work has been appreciated by the referees and very grateful for their suggestions.
In the following, I will reply point by point to the referee’s concerns as well as indicate all changes:

REPLY TO REPORT 1

1 - I revised the manuscript in order to correct as many spelling mistakes as possible,
and I removed a number of improper logical connectors. I hope now the readability of the manuscript is largely improved.

2 - I rephrased the first paragraph of the introduction as suggested by the referee, thus explicitly referring to the Ising model.
I slightly reformulated some passages of the introduction as well, in order to make the text more fluent.

3 - As pointed out by the referee, there is no connection with a S3 symmetry,
and nowhere in the paper I mentioned any sort of connection to S3.
However, there is no doubt that, in the stationary state, the generating function of the moments does coincide
with the partition function of a 1D classical model with 3 states;
it can be written as it was the 3-states Potts chain partition function, provided the classical couplings
are tuned in a very specific way (as carefully explained in the Appendix D).
Notwithstanding, I may understand that this wording could be “unhappy” thus leading to a possible misunderstanding;
therefore, I complied with the referee’s request and removed the word “Potts” from the text.
However, I do not think the Appendix D needs to be modified.

4 - Here the GGE has been used in order to exploit the extensive behaviour of the overlap with the ground state
in Eq. (78) and extract the scaling regime, namely the scaling of $\gamma$, and thereafter to get Eq. (82).
As pointed out by the referee, the Ground-State (GS) contribution to the stationary PDF has been already put forward in Ref. [33],
where a thermal stationary ensemble has been considered, since this phenomenon is not driven in any means by integrability.
However, in Ref. [33] only the large $\Delta$ expansion of the stationary PDF
has been performed (which is the counterpart of the small $h$ expansion here);
in addition, in this work, I considered the subsystem $\ell$ getting larger with $h\sqrt{\ell}$ kept fixed.
The GGE is crucial to extract in this regime the proper scaling of the stationary
PDF and connect it to both the ground-state PDF plus Gaussian fluctuations.
A couple of sentences have been added to this section to clarify this point.

REPLY TO REPORT 2

a) As suggested by both referees, the word “Potts” has been removed.
As well as, “proof” (spelling mistake) became “show”.

b) Typos corrected.

c) In order to avoid confusion, I decided to remove the local identity matrix, thus just keeping the operator notation,
where $(\sigma^{x}_{j})^2 = 1$.

d) Those calculation easily come from series expanding the Fourier coefficient in
Eq.s (34) (and (50)) and therefore integrating (i.e. Fourier transforming) term by term.
Now, after eq. (34), a sentence has been added to make this clear.
Regarding the notation of Eq.s (35), (36) and (51), as pointed out by the referee, it is finite when sin(pi * n) gets cancelled by the poles
of the series. A sentence has been added to point it out.

e) Maybe I did not fully understand the referee's comment: first, how much “remarkable” is the connection, in my very personal opinion,
is a matter of taste; second, has the referee perhaps forgotten that he suggested (point (a)) removing the word “Potts”?

f) I thank the referee for having pointed out the mistake: eq (69) comes from Szego theorem; the other from Fischer-Hartwig conjecture;
I also added proper references.

g1) As a matter of fact, the quantum Ising chain is a very well known model which has been solved more than fifty years ago.
Every times a section of the manuscript is based on some already known results (e.g. in Sec. 2, part of Sec. 3, few equations in Sec.s 4 and 5,
part of Sec. 6), I usually refer to them by citing the appropriate references at the very beginning.
My policy is to avoid to repeat every time in a while the same bibliographic citations.
Moreover, simply manipulation of well known formulae (e.g. Sec.3.1, Sec 3.2) as well as basic expansions (e.g. in Sec. 4.1, 5.1)
are such that they do not need any citation at all, independently whether this is the first time they have been written or not.

g2) The gaussian behaviour has been checked against the numerics (as provided by many figures in the paper),
and it is expected to be rigorously valid in the specific scaling regime explained in Sec. 3.1.
In other words, Eq. (44) is exact when $\ell\to\infty$ and $m$ is rescaled with $\sqrt\ell$.
The same in the ferromagnetic regime, when Gaussian restoration is mentioned after Eq. (48).
Similar reasoning applies for the Ground-State PDF.

h) There is no discussion because it is not special at all. The stationary generating function of the moment reduces
to the partition function of the classical Ising chain. In other words, the quench to $h=1$ is already included in the Sec. 4.1.1.
However, if the referee here is referring to the Ground-State properties at the critical point, well, the scaling of the
order parameter PDF has been already obtained by A. Lamacraft and P. Fendley in Ref. [56].

---

## Round 2 · List of Changes

see resubmission letter

---

## Editorial Decision

published